

# Ecomorphological analysis of the astragalo-calcaneal complex in rodents and inferences of locomotor behaviours in extinct rodent species

Samuel Ginot, Lionel Hautier, Laurent Marivaux and Monique Vianey-Liaud

Institut des Sciences de l'Evolution de Montpellier, Université de Montpellier, Montpellier, France

## ABSTRACT

Studies linking postcranial morphology with locomotion in mammals are common. However, such studies are mostly restricted to caviomorphs in rodents. We present here data from various families, belonging to the three main groups of rodents (Sciuroidea, Myodonta, and Ctenohystrica). The aim of this study is to define morphological indicators for the astragalus and calcaneus, which allow for inferences to be made about the locomotor behaviours in rodents. Several specimens were dissected and described to bridge the myology of the leg with the morphology of the bones of interest. Osteological characters were described, compared, mechanically interpreted, and correlated with a "functional sequence" comprising six categories linked to the lifestyle and locomotion (jumping, cursorial, generalist, fossorial, climber and semi-aquatic). Some character states are typical of some of these categories, especially arboreal climbers, fossorial and "cursorial-jumping" taxa. Such reliable characters might be used to infer locomotor behaviours in extinct species. Linear discriminant analyses (LDAs) were used on a wider sample of species and show that astragalar and calcaneal characters can be used to discriminate the categories among extant species whereas *a posteriori* inferences on extinct species should be examined with caution.

## INTRODUCTION

Rodents (Rodentia, Mammalia) are by far the most diverse and speciose mammalian order (e.g., *Wilson & Reeder, 2005*). Aside from this specific diversity, rodents occupy a wide array of ecological niches, from aquatic environments to desert areas, and this is notably reflected in the diversity of their locomotor and positional behaviours, which are associated with various types of locomotory apparatus. Most rodents are very versatile in terms of locomotion and are capable of using a variety of motions (e.g., jumping, running, climbing, or swimming). Highly specialized species are usually still able to switch from one type of locomotion to another (e.g., squirrels can climb a vertical support or run on a horizontal one, gerboas can use jumping running or quadrupedal walking). Other behaviours involving the limbs, such as digging and food handling, must also be considered since they influence the morphology as well as the modalities of locomotion (e.g., running

Corresponding author
Samuel Ginot,
samuel.ginot@umontpellier.fr

while carrying food or moving around in a burrow). Despite their geographical and taxonomic distance, some rodent communities converge in morphology and types of locomotion. One striking example is that of desert rodents, which display the same types of morphological adaptations on different continents, due to adaptive convergence linked to similar ecological and environmental constraints (*Eisenberg, 1975*).

The aforementioned behaviours, especially locomotor ones, are reflected in the post-cranial morphology. Although this relationship has been investigated in several mammalian groups, primarily in primates (e.g., *Gebo, 1988*; *Gebo, Dagosto & Rose, 1991*; *Strasser, 1992*) but also marsupials (e.g., *Szalay, 2006*), studies regarding rodents are still limited (e.g., *Szalay, 1985*; *Vianey-Liaud, Hautier & Marivaux, 2015*). Most have been focusing solely on caviomorphs (e.g., *Biknevicius, 1993*; *Weisbecker & Schmid, 2007*; *Lessa et al., 2008*; *Candela & Picasso, 2008*; *Morgan, 2009*; *Elissamburu & De Santis, 2011*), mostly because they display a wide array of ecological niches due to their long and isolated evolutionary history in South America (e.g., *Biknevicius, 1993*; *Weisbecker & Schmid, 2007*; *Lessa et al., 2008*; *Candela & Picasso, 2008*; *Morgan, 2009*; *Elissamburu & De Santis, 2011*). Two notable exceptions, combining numerous species from the three main clades of rodents with some fossil species, are found in *Samuels & Van Valkenburgh (2008)* and *Vianey-Liaud, Hautier & Marivaux (2015)*. Most of these articles focus on the girdles and/or limbs as a whole. However, one of the most important regions in locomotion, as well as other related behaviours, is the ankle joint, and chiefly the calcaneus and astragalus. Indeed, these two bones form the fulcrum (astragalus) and lever arm (calcaneus) of the foot, when taken as a lever system (*Carrano, 1997*; *Davidovits, 2012*). More interestingly, the calcaneus and astragalus are frequently found in the fossil record. Despite the pivotal roles of these two bones, and regardless the extreme diversity of the order, only a handful of studies has attempted to describe in detail the astragalo-calcaneal complex in rodents (e.g., *Szalay, 1985*; *Candela & Picasso, 2008*; *Vianey-Liaud, Hautier & Marivaux, 2015*).

This study aims to determine how locomotion imprints the morphology of the astragalus and calcaneus (especially the calcaneo-astragalar joint). This is crucial, since the shape of these two bones largely brackets the array of possible movements of the foot. In several taxa, comparable characters are described qualitatively, so that their variation can be functionally interpreted. This allowed us to define character states that best characterize the motion range of the ankle, and the favoured locomotor mode as a result. Legs were dissected in a few species in order to clarify the functional link between bones and muscles. Finally, we employed multivariate analyses on a larger dataset to infer the locomotory behaviours of some fossil species.

## MATERIAL AND METHODS

### Qualitative analyses

Precise anatomical descriptions were made on a total sample of 17 species in as many genera. These species were selected to represent a large panel of locomotor modes in several different families of rodents, to potentially detect convergent characters. To limit the bias of different stages of ossification, we used adult complete specimens or, when isolated

calcanei and astragali were studied, they were fully ossified. Osteological nomenclature was derived from *Szalay (1985)*. Classification follows *Fabre et al. (2012)*.

We sampled six already cleaned astragali and calcanei of six different species in six genera from the Muséum National d'Histoire Naturelle (MNHN) in Paris, France (Ctenohystrica: *Myocastor coypus* MNHN 1959-148; Squirrel-related clade: *Pteromys volans* MNHN 1929-433, *Marmota marmota* MNHN 1933-277, *Sciurus vulgaris* MNHN 2000-407; Myodonta: *Spalax ehrenbergi* MNHN 200-353, *Jaculus* sp. MNHN A12-495).

Eleven more astragali and calcanei of eleven other species in ten genera were sampled in the collections of the Université de Montpellier (UM), housed at the Institut des Sciences de l'Evolution de Montpellier (ISE-M), France (Ctenohystrica: *Proechimys cuvieri* UM 1054 V, *Cavia porcellus* UM 558 V, *Chinchilla lanigera* UM 498 V, *Octodon degus* UM 500 V, *Coendou prehensilis* N-481, *Ctenodactylus vali* UM 709 N; Squirrel-related clade: *Spermophilus fulvus* UM 280 N; Myodonta: *Psammomys obesus* UM 577 N, *Microtus gregalis* N-420, *Gerbillus* sp. UM 576 N, *Castor* sp. UM 055 V).

Six of these eleven specimens were already cleaned, the other five (*C. lanigera, C. vali, Gerbillus sp., P. cuvieri* and *P. obesus*) were complete animals fixed in formal saline (buffered 4% formaldehyde solution with 0.12 M NaCl) and stored in 70% ethanol. Their hindlimbs were dissected and cleaned to get the astragalus and calcaneum, as well as to clarify the links between these bones and the muscular and ligamentary systems of the leg at the level of the ankle. Since the myology of the leg in these species had not yet, to our knowledge, been studied, identifications of muscles, ligaments, and other structures during dissections was based on comparisons with reports by *Klingener (1964*; Dipodoidea), *Brannen* (*1979*; *Mus musculus*), and *Hildebrand (1978*; diverse taxa). In all dissections, one leg was cut off the specimen (except for *Proechimys cuvieri*, in which the state of conservation required dissecting both legs, without cutting them off) and some of the most proximal muscles were also cut in the process (e.g., *Biceps femoris* and *vastus lateralis*) and could not be described precisely. However, since the main interest here is the calcaneo-astragalar complex, we focused on muscles that show direct connection to these bones (*biceps femoris* and *vastus lateralis, gastrocnemius, soleus, tibialis cranialis* and *extensor digitorum longus, peroneus longus* and *brevis*, and *flexor fibularis*). For the five species that were dissected, the origin and insertion of the muscles connected to the astragalus and calcaneus are summed up in Table S1. The insertions of tendons and ligaments on the calcaneus and astragalus are summed up in Fig. 1. Once the legs were dissected, and the calcaneus and talus were cleaned of the remaining soft tissues, the osteological characters were studied qualitatively as with aforementioned specimens that were already cleaned. All abbreviations are summed up in Appendix 1.

## Locomotory categories

Following *Elissamburu & Vizcaíno (2004)*, a "functional sequence" is proposed, as follows: (1) "jumper" (*C. lanigera* and *Jaculus* sp.), includes species that use mainly hindlimb driven jumps to move around (either bipedal or quadrupedal); (2) "cursorial" (*Gerbillus* sp. and *M. coypus*), includes quadrupedal running taxa; (3) "generalist" (*P. cuvieri, C. gundi, C. porcellus* and *O. degu*), this category contains quadrupedal ground dwellers

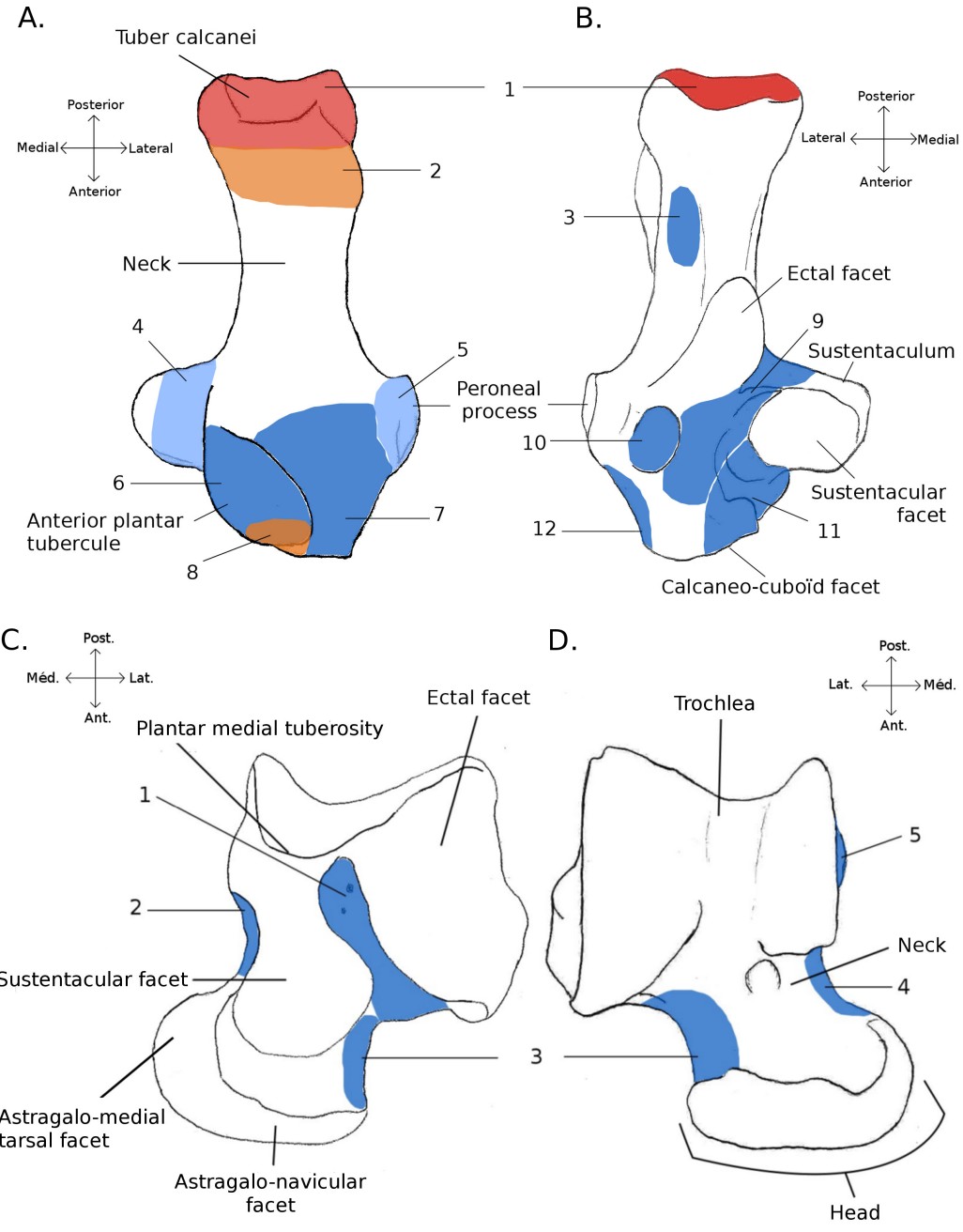

**Figure 1 Structures and insertions on the astragalus and calcaneus.** (A) Calcaneus of *Marmota marmota* (as an example) to show the main structures of the bone, in plantar (left) and dorsal (right) views. Numbers correspond to colored areas (red: tendon insertions; orange: tendon origins; light blue: passage of tendon; dark blue, ligament insertions. 1: insertion of Achilles tendon (*M. soleus* and *M. gastrocnemius*); 2: origin of *M. flexor digitorum brevis* (may be continuous with the insertion of the Achilles tendon); 3: concave area on which is inserted a ligament attached to the distal head of the crus; 4: area along which the tendons of *M. flexor fibularis* and *M. flexor digiti* slide (but do not attach); 5: area along which the tendons of *M. peroneus longus* and *brevis* slide; 6: area of insertion of ligaments linked to the cuboïd and metatarsal 4. On the lateral edge of the area is a ligament linked to the plantar side 

**Figure 1 (…continued)**
of the astragalus. 7: area of insertion of a ligament, generally linked to metatarsal 1. In *P. cuvieri* the area is concave and a small adductor muscle for digit 1 originates from there. 8: Origin of *M. flexor digiti V brevis*; 9: along this area are inserted fibers forming a slender ligament, linked to the plantar side of the astragalus. 10: concave area in which is inserted a ligament linked with the plantar side of the astragalus. 11: area of insertions of two ligaments, one linked with the neck of the astragalus (area 3 in B), the other with the navicular. Posteriorly, on the medial edge of this area, are attached ligaments linked with the medial tarsal bone and the ectocuneiform or mesocuneiform. 12: area of insertion of connective tissue linked to metatarsal 4. (B) Astragalus of *Marmota marmota* (as an example) in plantar (left) and dorsal (right) views. As in (A), areas of ligament insertions are represented. 1: *sulcus*, insertion of the ligament linked to the dorsal side of the calcaneus (9 in A). 2: area of insertion of connective tissue linking the astragalus with the medial malleolus and entocuneiform bone. 3: area of insertion of a ligament linked to the dorsal side of the calcaneus (11 in A). 4: area of insertion of connective tissue linked with the navicular and medial tarsal bones (see *Hildebrand, 1978* and *Szalay, 1985* for details on the medial tarsal bone). 5: area of insertion of the deltoïd ligament.

that are also know to commonly climb or dig burrows; (4) "fossorial" (*M. marmota, M. gregalis, P. obesus, S. fulvus* and *S. ehrenbergi*), includes adept diggers that are partly or fully fossorial; (5) "climber" (*S. vulgaris, P. volans,* and *C. prehensilis*), includes tree-dwelling taxa that are specialized in climbing. (6) "semi-aquatic" (*M. coypus* and *Castor* sp.), with taxa that spend a large part of their time in water, and are known to be good swimmers. The placement of each taxa in the different categories derives from the available literature dealing with the ecology and behaviours (categories and refs. are provided in Table 1). Qualitative characters were observed and correlated with their behaviours and with the sequence.

## Quantitative analyses

Fourty-two more specimens (representing 19 additional species) were sampled from the MNHN collection and used in Linear Discriminant Analyses (LDA), altogether with the taxa that were described in full details. Overall, 56 specimens in 35 species were used in the LDA. All individuals, as well as their locomotor group are listed in Table 1. For all individuals, 16 linear measurements were taken on the calcaneus and 22 on the astragalus (all measurements are shown in Fig. 2, and their values for each specimen are listed in Supplemental Information, Table 2). For the astragalus, however, some measurements presented in Fig. 2 (c1, c2, d1, d2, lTAH and mTAH) were not used in the following analysis, since they did not represent curvature precisely. Thus, we also used 16 measurements for the astragalus in our analysis. When several specimens were available for one species, the species' average for each measurement was calculated. Some of the measurements were taken from *Candela & Picasso (2008)*, while the rest were created to describe the morphological variation of the bones thoroughly, with regard to function, based on the qualitative characters. To remove the effect of size from the analysis, all linear measurements of each individual (or the average when several specimens were measured for one species) were divided by the corresponding geometric mean, and a log function was used to produce log shape ratios (*Claude, 2008*). The categories of the functional sequence were also used in these analyses. Calcanei and astragali were treated separately in order to check for differences in the signals between the astragalus and the calcaneus, as well

**Table 1  Summary of specimens studied.** Rodent specimens studied, their locomotor category and literature sources consulted. Fossil species are marked with an asterisk (*). Locomotory categories are numbered as follows: 1: "jumping"; 2: "cursorial"; 3: "terrestrial generalist"; 4: "fossorial/semi-fossorial"; 5: "arboreal climber"; 6: "semi-aquatic"; na designates fossil species, for which we try to assess the locomotion.

| Species | Collection number | Locomotory category | Ref. |
|---|---|---|---|
| *Anomalurus derbianus* | MNHN2003-150 | 5 | *Howell, Hutterer & Ekué (2008)* |
| *Atherurus africanus* | MNHN1992-1832 | 3 | *Hoffmann & Cox (2008)* |
| *Atherurus africanus* | MNHN1994-2450 | 3 | |
| *Blainvillimys langei** | UM | na | |
| *Capromys pilorides* | MNHN1881-1.004 | 5 | *Soy & Silva (2008)* |
| *Castor canadensis* | MNHN1996-520 | 6 | |
| *Castor canadensis* | MNHN1996-2168 | 6 | *Linzey et al.(2013)* |
| *Castor sp.* | UM055 V | 6 | |
| *Cavia porcellus* | UM558 V | 3 | *Cassini (1991)*, *Rood (1972)* and *Wilson & Geiger (2015)* |
| *Chinchilla lanigera* | UM498 V | 1 | *Spotorno et al. (2004)*, *Nowak (1999)* and *Wilson & Geiger (2015)* |
| *Coendou prehensilis* | UM481 N | 5 | *Eisenberg (1989)*, *Marinho-Filho, Queirolo & Emmons (2008)* and *Nowak (1999)* |
| *Coendou prehensilis* | MNHN1997-643 | 5 | |
| *Ctenodactylus gundi* | MNHN1963-940 | 3 | *Gouat & Gouat (1987)* |
| *Ctenodactylus vali* | UM709 N | 3 | |
| *Cuniculus paca* | MNHN1874-145 | 3 | *Queirolo, Vieira & Reid (2008)*, *Queirolo et al. (2008)* and *Wilson & Geiger (2015)* |
| *Cuniculus paca* | MNHN1923-1024 | 3 | |
| *Dasyprocta leporina* | MNHN2000-372 | 2 | |
| *Dasyprocta leporina* | MNHN1998-241 | 2 | *Emmons & Reid (2008)* and *Wilson & Geiger (2015)* |
| *Dasyprocta leporina* | MNHN1988-142 | 2 | |
| *Dasyprocta leporina* | MNHN2006-503 | 2 | |
| *Dasyprocta punctata* | MNHN1929-624 | 2 | *Ojeda et al. (2013)* and *Wilson & Geiger (2015)* |
| *Dasyprocta punctata* | MNHN1929-626 | 2 | |
| *Dolichotis patagonum* | MNHN2000-827 | 2 | |
| *Dolichotis patagonum* | MNHN1974-85 | 2 | *Ojeda & Pardinas (2008)* |
| *Dolichotis patagonum* | MNHN1982-152 | 2 | |
| *Dolichotis sp.* | MNHN1995-192 | 2 | |
| *Erethizon dorsatum* | MNHN1909-53 | 5 | *Weber (2004)* |
| *Eucricetodon atavus** | UM | na | |
| *Funisciurus anerythrus* | MNHN1998-2157 | 5 | *Grubb & Ekué (2008)* |
| *Gerbillus sp.* | UM576 N | 2 | *Blumberg-Feldman & Eilam (1995)* |
| *Hydrochoerus hydrochaeris* | MNHN2013-14 | 6 | *Queirolo, Vieira & Reid (2008)* and *Queirolo et al. (2008)* |
| *Hydrochoerus hydrochaeris* | MNHN2001-1972 | 6 | |
| *Hydrochoerus hydrochaeris* | MNHN1916-18 | 6 | |
| *Hystrix cristata* | MNHN1990-662 | 3 | *Grubb et al. (2008)* |
| *Hystrix cristata* | MNHN1922-386 | 3 | |
| *Issiodoromys pauffiensis** | UM | na | |
| *Jaculus sp.* | MNHNA12-495 | 1 | *Happold (1967)* and *Schröpfer, Klenner-Fringes & Naumer (1985)* |

**Table 1** (*continued*)

| Species | Collection number | Locomotory category | Ref. |
|---|---|---|---|
| *Lagidium peruanum* | MNHN1881-1218 | 2 | *Wund (2000)* |
| *Marmota marmota* | MNHN1996-2438 | 4 | *Barash (1976)*, *Mainini, Neuhaus & Ingold (1993)*, |
| *Marmota marmota* | MNHN1899-76 | 4 | *Perrin, Berre & Le (1993)* and *Herrero, García* |
| *Marmota marmota* | MNHN1933-277 | 4 | *González & García Serrano (2002)* |
| *Microtus gregalis* | UM420 N | 4 | *Batsaikhan et al. (2008)* and *Nowak (1999)* |
| *Myocastor coypus* | MNHN1959-148 | 6 | *Gosling (1993)*, *D'adamo et al. (2000)* and |
| *Myocastor coypus* | MNHN2000-395 | 6 | *Guichón et al. (2003)* |
| *Myocastor coypus* | MNHN1935-650 | 6 | |
| *Myoprocta acouchi* | MNHN2000-366 | 2 | *Catzeflis, Weksler & Bonvicino (2008)* and *Wilson & Geiger (2015)* |
| *Ocotdon degus* | UM500 V | 3 | *Woods & Boraker (1975)* and *Nowak (1999)* |
| *Palaeosciurus goti\** | UM | na | |
| *Pedetes capensis* | MNHNVI-1384 | 1 | *Jackson (2000)* |
| *Proechimys cuvieri* | UM1054 V | 3 | *Guillotin (1981)*, *Nowak (1999)* and *Wilson & Geiger (2015)* |
| *Psammomys obesus* | UM577 N | 4 | *Aulagnier & Granjon (2008)* |
| *Pseudoltinomys gaillardi\** | UM | na | |
| *Pteromys volans* | MNHN1929-433 | 5 | *Hanski et al. (2000)* |
| *Ratufa bicolor* | MNHNA.13-435 | 5 | *Walston, Duckworth & Molur (2008)* |
| *Ratufa indica* | MNHN1966-65 | 5 | *Rajamani, Molur & Nameer (2010)* |
| *Sciurus vulgaris* | MNHN2000-407 | 5 | *Lurz, Gurnell & Magris (2005)*, *Schmidt & Fischer* |
| *Sciurus vulgaris* | MNHN1992-1860 | 5 | *(2011)* and *Youlatos & Samaras (2011)* |
| *Spalax ehrenbergi* | MNHN200-353 | 4 | *Heth (1989)* and *Zuri & Terkel (1996)* |
| *Spermophilus fulvus* | UM280 N | 4 | *Çolak & Çolak (2007)*, *Tsytsulina, Formozov & Sheftel (2008)* and *Lagaria & Youlatos (2006)* |

as to allow us to use fossil specimens, for which bones are often found isolated. For both the astragalus and calcaneus, MANOVAs were run to test for effects of locomotion and phylogeny. Morphological variables that had no correlation with any linear discriminant functions were not used in the MANOVAs.

Fossil taxa from the UM collections were added *a posteriori* in the analyses (Theridomyidae: *Blainvillimys langei* RAV2001 and RAV2002, *Pseudoltinomys gaillardi* RAV2003 and RAV2004, *Issiodoromys pauffiensis* SPV593, SPV592 and MPF213; Sciuridae: *Palaeosciurus goti* MGB101 and MGB102; Cricetidae: *Eucricetodon atavus* RAV2005 and RAV2006). These specimens are all from French Oligocene deposits of the Quercy area (localities Ravet, Mas-de-Got-B and Mas-de-Pauffié). Taxon identification is based on faunal lists from these localities and comparison with material represented in *Vianey-Liaud, Hautier & Marivaux (2015)*. Post-cranial remains of *B. langei*, *P. gaillardi* and *I. Pauffiensis* are described in *Vianey-Liaud, Hautier & Marivaux (2015)* and details on *P. goti* can be found in *Vianey-Liaud (1974)*. Both these papers present hypotheses regarding the locomotion of these taxa. The specimens used here were selected to test these hypotheses, as well as to check how well can quantitative analyses predict locomotory behaviour in

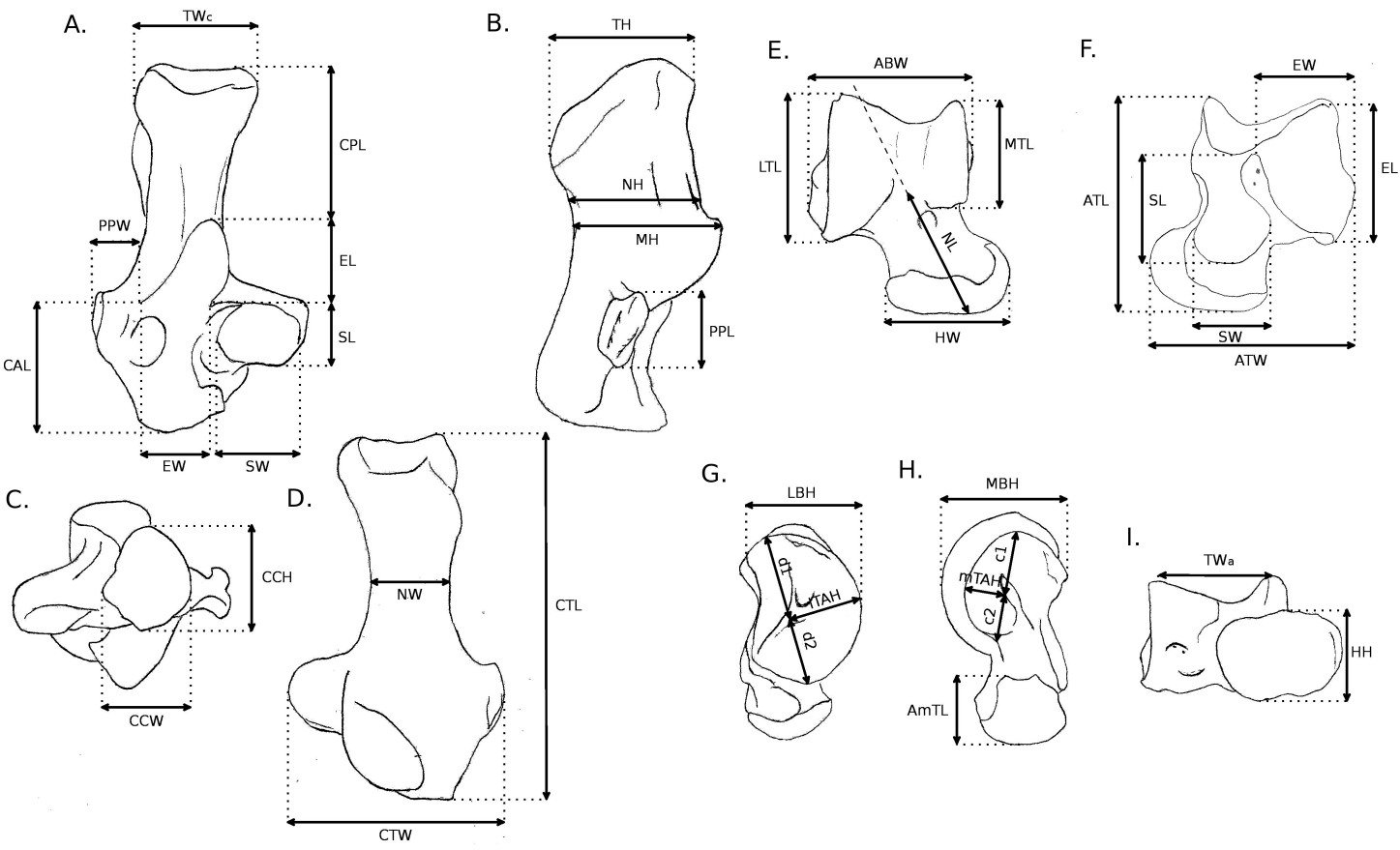

**Figure 2** **Measurements used in the study.** Linear measurements of the calcaneus and astragalus, abbreviations are given in Appendix 1.

**Table 2** **Structure matrix of the LDA of the astragalus.** Correlation coefficients of variables with the LD functions are presented. Significant correlations are marked in bold.

|  | LD1 | LD2 | LD3 | LD4 | LD5 |
|---|---|---|---|---|---|
| ABW | −0.10537176 | **−0.50098856** | 0.046093587 | 0.013015596 | **−0.46210440** |
| ATL | −0.11995258 | **0.36519710** | 0.164739754 | **−0.333163201** | **0.37761961** |
| ATW | 0.26204140 | −0.15665090 | 0.276320246 | **0.649285720** | −0.26499805 |
| EL | **0.32773828** | −0.02529528 | 0.261982032 | 0.191915531 | −0.32008916 |
| EW | −0.05817710 | −0.12669810 | −0.241478230 | −0.274803114 | −0.22041763 |
| HH | 0.16402700 | **0.45097531** | 0.303174467 | 0.011135147 | **−0.52531990** |
| HW | −0.04956075 | −0.01003520 | 0.270973137 | −0.110596893 | **0.52364075** |
| LBH | **−0.51423999** | −0.10762628 | 0.005310194 | −0.131664969 | **0.45065820** |
| LTL | −0.11572577 | 0.07809347 | **0.370167097** | **0.386093664** | −0.05059842 |
| MBH | **−0.57120785** | **−0.61366060** | −0.050626406 | 0.040909368 | 0.14008911 |
| MTL | **−0.65121417** | **−0.34532619** | −0.293433734 | 0.323860009 | 0.07402242 |
| NL | 0.29086529 | **0.39502449** | 0.234091001 | **−0.403318420** | 0.03408407 |
| SL | **−0.33267622** | 0.15018611 | **−0.673424330** | −0.143015501 | 0.23584098 |
| SW | **0.65460654** | −0.01896417 | 0.118607900 | **0.338424171** | **−0.32900799** |
| TWa | **0.54933895** | −0.10780003 | **−0.329979946** | −0.005278722 | 0.09038230 |
**Table 3  Structure matrix of the LDA of the calcaneus.** As in Fig. 2A.

|      | LD1        | LD2          | LD3         | LD4         | LD5         |
|------|------------|--------------|-------------|-------------|-------------|
| CAL  | **−0.6089226** | 0.009437009  | 0.26903032  | 0.30657590  | 0.01131940  |
| CCH  | **0.4365398**  | −0.225030318 | −0.06647538 | 0.21760746  | 0.21190580  |
| CCW  | **0.6846866**  | −0.071509066 | −0.23823943 | −0.04327920 | −0.18043648 |
| CPL  | **−0.3832592** | 0.180004165  | 0.24599583  | −0.32329036 | **−0.35542347** |
| CTL  | **−0.6537200** | −0.030370030 | −0.02745715 | **0.36510847**  | 0.09558776  |
| CTW  | **0.7936011**  | 0.010207707  | **−0.35342291** | 0.24844656  | 0.21476611  |
| EL   | 0.1632429  | 0.153461556  | **0.44157589**  | −0.16975778 | −0.02279185 |
| EW   | −0.1930453 | 0.053702045  | 0.21899552  | **−0.36521510** | **−0.46257422** |
| MH   | **−0.6371164** | 0.216593121  | −0.04388484 | 0.04501014  | 0.32716889  |
| NH   | −0.1822259 | 0.295394822  | 0.30618512  | −0.17328258 | −0.02273043 |
| NW   | −0.2464843 | **−0.379262339** | −0.30127783 | −0.11361849 | **0.58602078**  |
| PPL  | 0.3166758  | −0.163289434 | 0.06851816  | 0.28356598  | −0.18724439 |
| SL   | 0.0306049  | **0.473558898**  | **−0.34657326** | −0.27026256 | −0.16754147 |
| SW   | **0.3473685**  | −0.165810698 | 0.08090421  | −0.22729684 | 0.28674243  |
| TH   | **0.4020693**  | 0.087112252  | −0.05230598 | 0.18863164  | 0.09423960  |
| TWc  | **−0.3826445** | **−0.479335954** | **−0.37882048** | 0.29881183  | 0.16442349  |

fossil group with extant representatives (*P. goti, E. atavus*) or not (*B. langei, P. gaillardi* and *I. Pauffiensis).*

For a more synthetic view, all the specimens (extant and extinct) are listed in Table 1. All analyses were performed with R (*R Core Development Team, 2015*), using the function lda of package MASS (*Venables & Ripley, 2002*).

# RESULTS
## Osteological descriptions
This part includes descriptions of the astragali and calcanei of four species in three families from the Myodonta (Muridae, Cricetidae and Spalacidae), six species in five families from Ctenohystrica (Ctenodactylidae, Cavidae, Erethizontidae, Octodontidae and Chinchillidae) and four species in one family from the squirrel-related clade (Sciuridae) (*Fabre et al., 2012*). In cases where several species were described in the same family, only one comparative description was made. When only one species was available for a said family, this species was described in full details.

**Sciuridae** Gray, 1821 (Figs. 3A–3D)

Four sciurid representatives were considered: *Marmota marmota* Linnaeus, 1758 (Fig. 3A) and *Spermophilus fulvus* Lichtenstein, 1823 (Fig. 3B) (both terrestrial squirrels from the Xerinae), and *Sciurus vulgaris* Linnaeus, 1758 (Fig. 3D) and *Pteromys volans* Linnaeus, 1758 (Fig. 3C) (both tree-dwelling squirrels from the Sciurinae).
**Calcaneus.** The ectal facet of the calcaneus is generally long and helical, but much less so in the alpine marmot. The sustentacular facet has a round shape in ground species, while

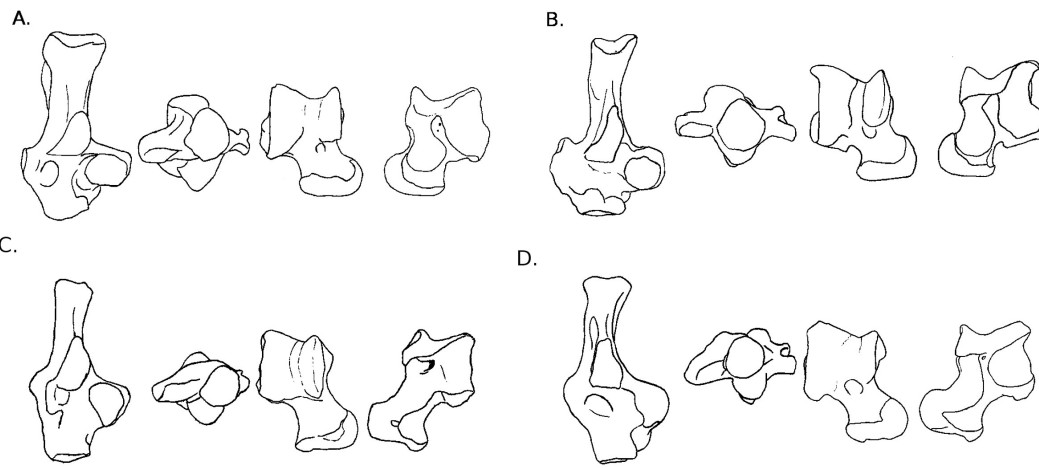

**Figure 3** **Drawings of calcanei and astragali of squirrel related species.** Calcanei (dorsal and anterior views) and astragali (dorsal and plantar views) of squirrel-related clade specimens described in Results—Osteological descriptions—(A) *Marmota marmota*; (B) *Spermophilus fulvus*; (C) *Pteromys volans*; (D) *Sciurus vulgaris*.

it is rather triangular in climbers, furthermore it is slightly concave in all species except *M. marmota*. Both facets are always separated by a distinct sulcus (Fig. 1A). In all species, the peroneal process is well-developed and in a posterior position. This is particularly true in climbing squirrels (*P. volans* and *S. vulgaris*), in which the peroneal process is even more posterior than the sustentacular process (also noted in *Emry & Thorington, 1982* and *Thorington Jr et al., 2005*). The position of the peroneal process and the importance of its groove are linked to the various movements of the *peroneus* muscles tendons (Fig. 1). The neck of the calcaneus is long and narrow, and curved medially, with the exception of the marmot, in which it is wider and less curved. The tuber calcanei show to distinct crests. In all species, except *M. marmota*, the medial crest is more developed than the lateral one, and it slightly projects medially. The body is rather short in the two ground squirrels, and longer in the climbers, especially *P. volans*. In plantar view, the anterior plantar tubercle is well-developed in the ground squirrels only. The groove for the *flexor fibularis* muscle (Fig. 1) is deeper and wider in ground squirrels than in climbers. Finally, the calcaneo-cuboid facet is generally round and concave in all species, but it is slightly more ovoid in the alpine marmot.

**Astragalus**. The sides of the astragalar trochlea are always asymmetrical, but more so in *S. vulgaris* and *P. volans*. The groove of the trochlea is also shallower and narrower in these latter species, compared to the ground squirrels. The neck is generally long and it deviates medially, particularly in climbers. The head of the astragalus is rather wide and round, but the astragalo-navicular facet is saddle-shaped in both *S. vulgaris* and *P. volans* (contrary to *Emry & Thorington, 1982*, who restricted this character to the Sciurini). The AmT facet is generally well-developed on the medial side of the neck, but is also more developed on the dorsal side of the neck in climbers. On the plantar side, the sustentacular and ectal facets are oriented mainly antero-posteriorly in the ground squirrels, while they are obliquely oriented in the tree squirrels. Furthermore, a sulcus is always present between the facets

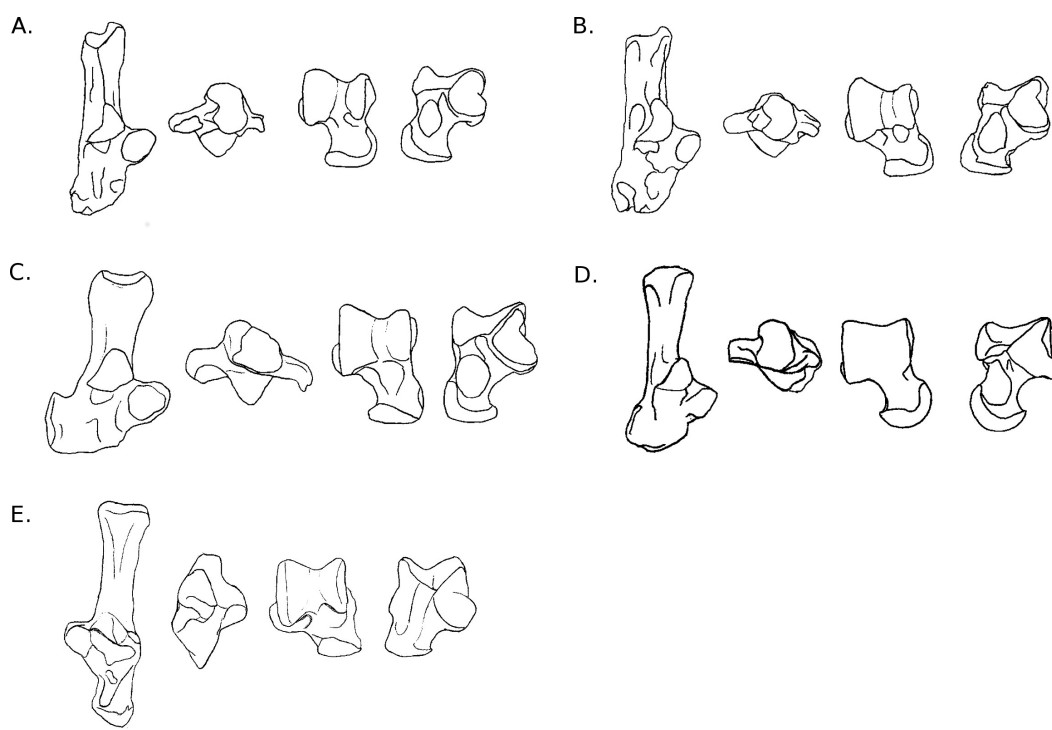

**Figure 4 Drawings of calcanei and astragali of Myodonta species.** Calcanei (dorsal and anterior views) and astragali (dorsal and plantar views) of Myodonta specimens described in Results—Osteological descriptions—(E) *Gerbillus sp.*; (F) *Psammomys obesus*; (G) *Microtus gregalis*; (H) *Spalax ehrenbergi*; (I) *Jaculus sp.*

(Fig. 1B), except in *P. volans*. The sustentacular facet joins the AN facet in the climbers, while they are separated in the ground squirrels. Although the medial plantar tuberosity is developed in all species, it is very prominent and forms a "hook" in the tree squirrels. On the medial and lateral sides, it appears that the sides of the trochlea are more curved in ground squirrels than in tree squirrels.

**Muridae** Illiger, 1811. (Figs. 4E–4F)

Two murid representatives are described: *Gerbillus* sp. Desmarest, 1804 (Fig. 4E), and *Psammomys obesus* Cretzschmar, 1828 (Fig. 4F). Both are members of the Gerbillini tribe in the Gerbillinae subfamily.
**Calcaneus**. The ectal facet of the calcaneus is strongly convex in both species, but longer in *P. obesus*. Both species show an ovoid sustentacular facet, slightly smaller in *P. obesus*. The neck of the calcaneus is straight in both species, and narrower and longer in *Gerbillus* sp. In both species, the tuber shows a well-developed medial posterior crest, while the lateral side is flatter. *P. obesus* also shows a shallow fossa on the medial side of the neck, and a deep and ovoid one on the lateral side, allowing strong attachment of the ligament linking the calcaneus to the crus. The body is elongated, and slightly widened in *P. obesus*. In both species the calcaneo-cuboid facet is large and bean-shaped. In *Gerbillus* sp. this facet is obliquely oriented, the lateral edge being more distal than the medial edge. Differences are

notable in plantar view in *Gerbillus* sp. The groove for the *flexor fibularis* muscle is deep, while it is shallow in *P. obesus*. Furthermore, the anterior plantar tubercle is well-developed and projected anteriorly in *Gerbillus* sp. while it is rather weak in *P. obesus*. The peroneal process is not developed in *Gerbillus* sp. and only slightly more in *P. obesus*. However, in both cases it bears a well-defined groove for the passage of the *peroneus* muscles. In both species, the peroneal process is placed very distally on the body of the calcaneus.

**Astragalus**. The trochlea is asymmetrical and very wide in both species. The groove between sides is shallow, with *Gerbillus* sp. being slightly deeper. Both species show a pit at the base of the medial ridge of the trochlea. The neck is wide and moderately long in both species, however it is only deflected medially in *P. obesus*. The head is wide and the AN facet is developed on the dorso-lateral part of the neck. Similarly, the AmT facet is well-developed on the lateral side of the neck. In plantar view, the ectal facet is wide in both species, but projects more laterally in *Gerbillus* sp. The sustentacular facet is small, ovoid and oriented antero-posteriorly in both species. The ectal and sustentacular facets are separated by a wide sulcus, and so are the sustentacular and AN facets. In both species, the body is deep, with curved sides of the trochlea. In *Gerbillus* sp. the curvature is greater on the lateral side, while the reverse is true for *P. obesus*.

**Cricetidae** Fischer, 1817. (Fig. 4G)
*Microtus gregalis* (Arvicolinae, Arvicolini) Pallas, 1779. (Fig. 4G)

**Calcaneus**. The ectal facet of the calcaneus is short, convex and oriented medially. The sustentacular facet is ovoid in shape, elongated on an anteromedial to posterolateral axis. The sustentaculum is projected in its posterior part. The body and neck are thick, the neck widening posteriorly. The medial posterior crest of the tuber is salient, contrary to the lateral one. The peroneal process in *M. gregalis* has a peculiar shape compared to other taxa studied here, being strongly developed laterally and elongated anteroposteriorly, with a dorsal crest projected in its most posterior part. It is extremely distal, the edge being more distal than the surface of the calcaneo-cuboid face, while the posterior extremity is more distal than the fossa at the base of the ectal facet. The lateral groove of the process is wide and well-marked. The calcaneo-cuboid facet is enlarged transversely and in its plantar part. In plantar view, the anterior tubercle is small, but the groove for the *flexor fibularis* is present on the lateral part of the sustentaculum.

**Astragalus**. The astragalar trochlea is asymmetrical and the groove between the rims is shallow. The neck is moderately long and very slightly deflected medially. There is a well-defined protuberance in the middle of its dorsal aspect (probably a ''tibial-stop''). The astragalo-navicular facet is slightly convex, and its lateral edge is projected laterally. The AmT facet is not visible in dorsal view. On the plantar side, the ectal facet is long and has a crescent shape. It is slightly oriented towards the lateral side, and the distal part is projected laterally; it is also strongly curved. The sustentacular facet is ovoid, oriented anteroposteriorly, and clearly separated from the other plantar facets by a very wide and deep sulcus. The proximo-plantar edge of the trochlea is salient, but only where it joins the ectal facet. The medial tuberosity is rounded. The AmT facet can be seen in plantar view;

it is narrow and elongated on the medial part of the neck. In lateral and medial views, the trochlea displays a very low radius of curvature, and its body is deep.

**Spalacidae** Gray, 1821. (Fig. 4H)
*Spalax ehrenbergi* (Spalacinae) Nehring, 1898. (Fig. 4H)

**Calcaneus.** The neck is long when compared to the body; it is narrow but deep, especially in its posterior part. On the plantar side and anterior to the tuber, a clear protuberance allows the insertion for the short plantar muscles (Table S1). The ectal facet is short, convex, and entirely oriented medially. The sustentacular facet is reduced and clearly curved medially. The posterior edge of the sustentaculum joins the edge of the ectal facet. The groove between the ectal and sustentacular facets is weakly marked. The peroneal process is well-developed and distal-most. The anterior plantar tubercle is developed, medially positioned, and meets the edge of the calcaneo-cuboid facet. The groove for the *flexor fibularis* is wide but moderately deep. These characters are consistent with the development of a powerful musculature for the movements of the foot.

**Astragalus.** The astragalar trochlea is symmetrical; the groove is almost non-existent (flat trochlear surface). The neck is long, but only slightly deflected medially, and the head is large and spherical. On the plantar side, the ectal facet is reduced in its distal medial part and somewhat curved. The sustentacular facet is short and oriented anteroposteriorly, although it is prolonged medially in its distal part, therefore joining the AmT facet. The sulcus between the ectal and sustentacular facets is short and shallow, but wide. The trochlea is moderately curved on both sides, and its body is deep.

**Dipodidae** Waterhouse, 1842. (Fig. 4I)
*Jaculus* **sp.** (Dipodinae) Erxleben, 1777. (Fig. 4I)

**Calcaneus.** On the calcaneus, the ectal and sustentacular facets are concave and oriented almost perpendicularly to the major axis of the bone. Therefore, the astragalus interlocks into these facets, which limits or potentially stops any movement of one bone independently of the other. The body of the calcaneus and the calcaneo-cuboid facet are narrow and oriented dorsoplantarly. The neck is long and moderately deep. The sustentaculum and the medial tarsal form a continuous surface, which strongly maintains the astragalus in place. The groove for the *flexor fibularis* is well-marked. The peroneal process is reduced to a weak groove, reflecting the reduction of the fibula, almost fused with the tibia.

**Astragalus.** Both sides of the astragalar trochlea are only slightly asymmetrical, and the groove between them is deep, thereby limiting movements in a parasagittal plane. Just distally to the base of the trochlea are two pits, which correspond to the contact with the anterior distal edge of the tibia during dorsiflexion of the foot. The neck is extremely short, almost non-existent, and the head is narrow and in line with main axis of the bone. On the plantar side, the sustentacular facet is oriented anteroposteriorly, extending from the plantar edge of the trochlea to the astragalo-navicular facet. The ectal facet also has a particular shape: it is strongly curved and occupies the whole lateral side of the astragalus.

Its anterior part is also strongly projected laterally. During the joint movements, this lateral astragalar process remains in contact with the base of the calcaneal ectal facet. In lateral view, the trochlea is curved; it is slightly less in medial view.

In this taxon it seems that both bones form one tightly fitted unit, with very little independent movements from one another.

**Ctenodactylidae** Gervais, 1853. (Fig. 5J)
*Ctenodactylus  vali* Rothmann, 1776. (Fig. 5J)

**Calcaneus**. The ectal facet of the calcaneus is short, convex, and in its anterior lateral extremity is projected laterally. The sustentacular facet is small with a slightly concave anterior part. The two facets are merged, without apparent limit or groove. Furthermore, the body of the bone is wide, suggesting an efficient support as well as mobility for the astragalus. The peroneal process is moderately developed, distally positioned and bears a well-defined groove; as a whole it looks similar to that of *Coendou prehensilis*.
**Astragalus**. The trochlea is asymmetrical and only slightly grooved. The neck is long (but less than in squirrels), and clearly deflected medially. The head is wide and rounded in its medial part. On the plantar side, the ectal and sustentacular facets are merged together, forming a smooth surface transversely oriented. The ectal part of the surface is somewhat curved, especially posteriorly. The articular surface of the sustentacular part is developed on the neck but does not reach the astragalo-navicular facet. The plantar medial tuberosity is prominent and hook-shaped, which articulates with the posterior edge of the sustentaculum. In medial and lateral views, the trochlea is moderately curved, and the body is deep.

**Caviidae** Waterhouse, 1829. (Fig. 5K)
*Cavia porcellus* (Caviinae) Linnaeus, 1758. (Fig. 5K)

**Calcaneus**. The ectal facet of the calcaneus is convex and short. The surface of the sustentacular facet is parallel to the major axis of the bone. Both facets are in contact, although the limit between them can easily be identified. The neck is straight, narrow and deep. The body of the bone is oriented dorso-plantarly, as is the calcaneo-cuboid facet. This facet is also crescent-shaped, and its dorsal tip is more anterior than the plantar one. It is somewhat similar to what can be observed in some artiodactyls such as cows or goats. The groove for the *flexor fibularis* is deep, in relation with powerful toe flexing muscles. The peroneal process is absent and only a weak groove can be found on the anterior lateral side of the bone.
**Astragalus**. The astragalar trochlea is slightly asymmetrical, with a wide but moderately deep groove. The neck is very short and barely deflected medially. At the base of the central part of the trochlea, a deep and wide fossa allows the articulation with the anterior distal process of the tibia during dorsiflexion of the foot. The astragalo-navicular extends dorsally and projects laterally. In dorsal view, only the medial edge of the AmT facet is visible. In plantar view, the ectal facet is strongly concave and wide, in particular in its central part.

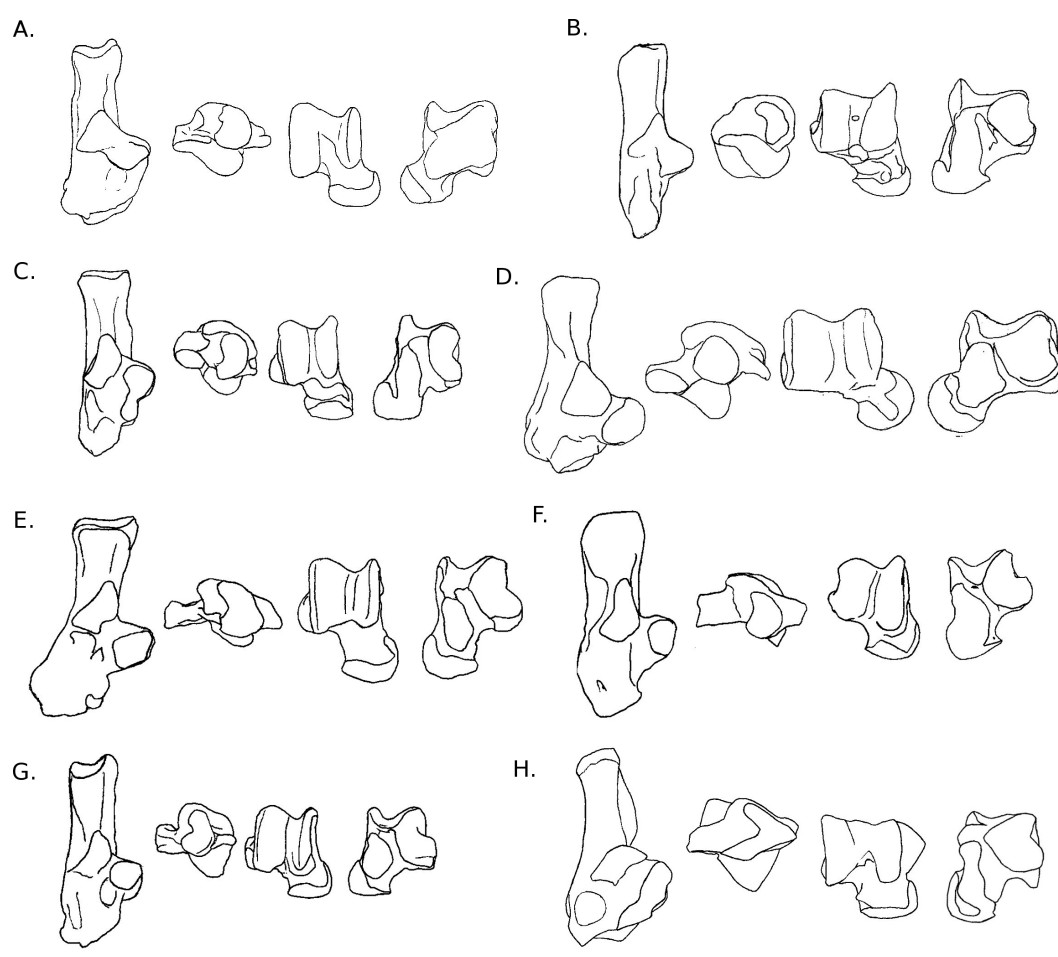

**Figure 5 Drawings of calcanei and astragali of Ctenohystrica species.** Calcanei (dorsal and anterior views) and astragali (dorsal and plantar views) of Ctenohystrica and Castoridae specimens described in Results—Osteological descriptions—(J) *Ctenodactylus* vali; (K) *Cavia porcellus*; (L) *Chinchilla lanigera*; (M) *Coendou prehensilis*; (N) *Octodon degus*; (O) *Myocastor coypus*; (P) *Proechimys cuvieri*; (Q) *Castor* sp.

It is also noticeably projected laterally in its anterior part. The sustentacular facet is long, joining distally the astragalo-navicular facet. It is separated from the ectal facet by a wide sulcus. The proximo-plantar edge of the trochlea is not salient, except at the level of the ectal facet, and there is no medial tuberosity. The AmT is short and clearly seen in medial and plantar view. The trochlea is slightly curved on both sides.

**Chinchillidae** Bennett, 1833. (Fig. 5L)
*Chinchilla lanigera* Molina, 1782. (Fig. 5L)

**Calcaneus.** The ectal facet of the calcaneus is convex and moderately long. The sustentacular facet is concave and extended anteriorly. The body and neck of the bone are aligned and similar in width. The calcaneo-cuboid facet is very oblique, turned towards the sustentaculum medially. In anterior view, it is oriented anteroposteriorly and crescent-shaped. In plantar view, the groove for the *flexor fibularis* is narrow and moderately deep.

The anterior plantar tubercle is marked and splits anteriorly into two protuberances of similar size. The peroneal process is reduced to a weak groove at the distal extremity of the lateral side of the bone.

**Astragalus**. The astragalar trochlea is slightly asymmetrical, with a deep groove between the two salient ridges. The neck is short and barely deflected medially. The astragalo-navicular facet is developed on the dorsal aspect of the neck, while the AmT facet is not visible. The dorsal aspect of the neck also bears a narrow transverse protuberance extending on the whole width of the neck, which is probably a tibia stop. On the plantar side, both the ectal and sustentacular facets are elongated anteroposteriorly, and separated by a deep and wide sulcus. The sustentacular facet extends all along the neck and joins the astragalo-navicular facet. The ectal facet is projected laterally and is strongly concave. The AmT facet is visible and developed on the anterior part of the plantar side of the neck. Both sides of the trochlea are moderately curved. Furthermore, in lateral view, the ectal facet occupies most of the lateral side of the body.

> **Erethizontidae** Bonaparte, 1854. (Fig. 5M)
> *Coendou prehensilis* (Erethizontinae) Linnaeus, 1758. (Fig. 5M)

**Calcaneus**. The ectal facet of the calcaneus is short but wide, convex and completely facing towards the medial side. The sustentacular facet is longer than wide and slightly concave. Both facets barely join, but are still separated by a thin but marked groove. The neck is stout and widens posteriorly. It is slightly curved towards the medial side (reminiscent of *Sciurus* and *Pteromys*) and is particularly tall in medial or lateral view. The medial crest of the tuber is salient and projected medially. The body is short and wide. The peroneal process is well-developed laterally, adding to the width of the body, but also anteroposteriorly. It is in a distal position and bears a marked groove, visible in plantar view. The calcaneo-cuboid facet is somewhat oblique in dorsal view; its lateral edge is more anterior than the medial one. In anterior view, it is concave, with an almost circular shape, widening medially. On the plantar side, the medial distal part of the neck bears a strong protuberance, on which Achilles tendon inserts and the short flexor muscle of the toes originates. The anterior plantar tubercle is marked, but does not protrude much. The groove for the *flexor fibularis* is moderately deep, but very wide.

**Astragalus**. The body of the astragalus is much wider than long. Both rims of the trochlea are asymmetrical, with a wide and moderately deep groove in-between. The neck is short and strongly deflected medially. The head is spherical, and the AmT and astragalo-navicular facets join and form a circular surface. In plantar view, the ectal facet is wide and long, but only slightly curved. The sustentacular facet is long, wide and oriented towards the medial side, extending on the neck and barely joining the astragalo-navicular facet. The sulcus between the ectal and sustentacular facets is marked, but short and narrow, probably allowing important transverse movements. The plantar edge of the trochlea is not salient and there is no clear medial tuberosity. The medial side of the trochlea is somewhat curved, but less than the lateral side. The body is tall on both sides. Overall, the astragalus of *C. prehensilis* is quite typical of erethizontid morphology (*Candela & Picasso, 2008*).

**Octodontidae** Waterhouse, 1839. (Fig. 5N)
*Octodon degus* Molina, 1782. (Fig. 5N)

**Calcaneus**. The ectal facet of the calcaneus is long and weakly convex; its medial edge is delineated by a marked groove. The sustentacular facet is medium-sized, ovoid (the major axis being oriented anterolateral to posteromedial), and almost flat. Both facets are widely separated. The neck is short and widens posteriorly. The medial posterior crest of the tuber is slightly more developed than the lateral one. The body is long and relatively flat (with the exception of a sub-ectal fossa) and widens anteriorly. The peroneal process is developed and occupies a distal position. It bears a shallow groove, which is visible in plantar view. The calcaneo-cuboid facet is slightly turned towards the sustentaculum medially. In anterior view, it has a crescent shape, with a noticeable enlargement in its dorsal part. The groove for the *flexor fibularis* is wide but moderately marked. The anterior plantar protuberance is not developed.

**Astragalus**. The astragalar trochlea is slightly asymmetrical, with a deep groove. The neck is quite long and displaced medially. The astragalo-navicular facet is developed on the dorsal part of the neck, in particular laterally. In plantar view, the ectal facet is clearly curved, longer than wide, but with an important lateral projection. The sustentacular facet is somewhat oval-shaped and oriented anteroposteriorly. It is separated from the ectal facet by a deep and wide sulcus, and does not join the astragalo-navicular or AmT facet. The plantar edge of the trochlea is not salient, with a small medial tuberosity. In lateral and medial views, the trochlea is quite curved, and the body is deep.

**Echimyidae** Gray, 1825. (Figs. 5O–5P)
*Myocastor coypus* Molina, 1782. (Fig. 5O)

**Calcaneus**. The calcaneus has a very unique morphology. The bone is massive, with a compact neck and tuber. The ectal facet is wide, long, and almost parallel to the medial side of the neck. It is concave at its base, but becomes convex in its posterior part. The sustentacular facet is slightly concave and parallel to the main axis of the bone. Both facets are widely separated by a groove. The groove for the *flexor fibularis* is also very deep, suggesting this muscle is strong. The peroneal process is rather reduced. The calcaneo-cuboid facet is distally positioned onto the long body, obliquely oriented towards the medial side. The anterior plantar tubercle is not prominent.

**Astragalus**. The astragalar trochlea is slightly asymmetrical and deeply grooved. At the base of the medial side is a cotylar fossa, distally bordered by an osseous bead, which stops the tibia during dorsiflexion of the foot. The neck is very short and slightly defected medially. On the plantar side, the sustentacular facet is wide and oriented anteroposteriorly; it meets the AmT facet medially. The ectal facet is very wide and projects laterally in its anterior part. It is also weakly curved, oriented obliquely, and faces the lateral side, which corresponds to the orientation of the calcaneal ectal facet. The posterior edge of the ectal facet directly meets the plantar edge of the trochlea, which does not overhang the facets. There is a deep,

long and oblique (posteromedial-anterolateral) sulcus between the ectal and sustentacular facets. Both sides of the trochlea are weakly curved, and its body is tall.

**Proechimys cuvieri** Petter, 1978. (Fig. 5P)

**Calcaneus.** The ectal facet of the calcaneus is short and convex, its posterior part turns medially. The sustentacular facet is small, circular and slightly concave. The sustentaculum is, however, well-developed and projected anteriorly. Both facets are separated by a wide groove. The body and neck are of similar widths. The posterior medial crest of the tuber is very salient, contrary to the lateral one. The body is longer than wide. In dorsal view, the calcaneo-cuboid facet is very oblique, turning towards the medial side. In anterior view, it is deep but narrow, and bean-shaped. The peroneal process is extremely distal and almost joins the lateral edge of the calcaneo-cuboid facet. It is small, but clearly recognizable, forms a half-circle on the lateral side of the bone, and bears a wide but shallow groove. In plantar view, the lateral distal part of the bone is concave, a condition which is found only in this taxon. According to the dissection of the specimen (see Table S1), a short muscle originates in this concavity, which inserts on the tendon of the first toe flexor muscle. Considering the direction and insertion of this muscle, it probably is an adductor for the first toe. The anterior plantar tubercle is weak, but the groove for the *flexor fibularis* is well-marked in a lateral position on the plantar side of the sustentaculum.

**Astragalus.** The astragalar trochlea is slightly asymmetrical and only slightly grooved. The medial rim is curved medially in its posterior part. The neck is short and slightly deflected medially. The astragalo-navicular is slightly convex and extends on the dorsal aspect of the neck. The AmT facet is limited on the dorsal side of the neck, but broadly developed on the plantar and medial sides. On the plantar side, the ectal facet is long, wide and curved, and its anterior part projects laterally. The proximo-plantar edge of the trochlea is salient but there is no clear medial tuberosity. The sustentacular facet is small, ovoid and oriented anteroposteriorly. It is separated from the ectal facet by a deep and very wide sulcus, and almost in contact with the AmT facet. The latter is narrow and elongated on the plantar side of the neck. In lateral view, the ectal facet occupies most of the side of the astragalar body, which is tall. The lateral and medial sides are strongly curved.

**Remark.** Although members of the same family, *P. cuvieri* and *M. coypus* do not exhibit many resemblances; the only shared characters are the lateral projection of the astragalar ectal facet and the development of the astragalo-navicular facet on the dorsal side of the neck. However, they are also shared with more distant taxa (e.g., *O. degus*, *C. lanigera*, *Jaculus sp.*), and probably reflect similar constraints during movements of the foot. This morphological heterogeneity can be explained by the uniqueness of the astragalus and calcaneus of *M. coypus*.

**Castoridae** (Fig. 5Q)
  *Castor* **sp.** (Fig. 5Q)

**Calcaneus.** The ectal facet is short, oriented towards the medial side, and projected plantarly. The sustentacular facet is particularly long and slender, concave, and it is oriented obliquely.

Both facets are separated by a deep and wide sulcus. The sustentacular process is projected plantarly. The neck is slender, but very developed dorso-plantarly. In particular, the plantar side forms a strong ridge that is projected medially. The tuber does not show developed medial or lateral ridges, however, there is a marked fossa at the insertion of the Achilles tendon (Fig. 1). The body is rather short, with an large fossa on the lateral side. The lateral part of the body is also projected distally. The calcaneo-cuboid facet has a left-arrow shape. In dorsal view, it is oriented obliquely and in line with the sustentacular facet. The groove for the *flexor fibularis* (Fig. 1) is deep and wide. The anterior plantar tubercle is present but not prominent. The peroneal process is developed and very distally placed. It bears a wide groove for the tendons of the *peroneus* muscles.

**Astragalus**. The trochlea of the astragalus is very wide and asymmetrical, the lateral side being larger than the medial side. However, the groove between the two sides is very shallow. The neck is short and wide; it is also slightly deviated medially. It bears a short ridge on the lateral side. The head is wide and the AN facet is convex and ovoid in shape. The AmT facet is long and comes all the way along the medial side of the neck. On the plantar side, the ectal facet is very wide, but not much curved. The sustentacular is very long, with an oblique medio-proximal to latero-distal axis. It joins with the AN facet on the lateral plantar side of the head. The ectal and sustentacular facets are separated by a wide and deep sulcus. The medial part of the astragalar body forms a process not seen in any other species studied here. The proximal extremity of the sustentacular facet is placed on the plantar side of this process. The body of the astragalus is not deep, and the sides of the trochlea have medium curvature.

## Quantitative analyses

The linear discriminant analyses (LDA) succeeded at separating the different locomotor groups, with some differences between the astragalus (Fig. 6) and calcaneus (Fig. 7). The effects of locomotion and phylogeny were significant in the analyses. About 40% of the taxa were assigned to their correct locomotor group when using leave-one-out cross validation.

In both the calcaneus and the astragalus analyses (Figs. 6 and 7), the three jumping taxa (*Chinchilla lanigera, Jaculus sp.,* and *Pedetes capensis*) are always found apart from all other categories. The climbers (*Anomalurus derbianus, Capromys pilorides, Coendou prehensilis, Erethizon dorsatum, Funisciurus anerythrus, Pteromys volans, Ratufa bicolor, Ratufa indica,* and *Sciurus vulgaris*) and fossorial/semi-fossorial (*Marmota marmota, Microtus gregalis, Psammomys obesus, Spalax ehrenbergi*, and *Spermophilus fulvus*) taxa are well-discriminated from cursorial (*Cavia porcellus, Dolichotis patagonum, Dolichotis sp., Gerbillus sp., Lagidium peruanum,* and *Proechimys cuvieri*), generalist (*Atherurus africanus, Ctenodactylus gundi, Ctenodactylus vali, Cuniculus paca, Dasyprocta leporina, Dasyprocta punctata, Hystrix cristata, Myoprocta acouchi, Octodon degus)* and semi-aquatic ones (*Castor* sp.*, Hydrochoerus hydrochaeris,* and *Myocastor coypus*). The coypu is, however, close to semi-fossorial species based on the astragalus (Fig. 6), this is not completely surprising, since this species is known to dig burrows several meters long. Cursorial taxa are confounded with terrestrial generalists in the astragalus analysis (Fig. 6). For the calcaneus (Fig. 7), however, both groups are quite well discriminated. The semi-aquatic

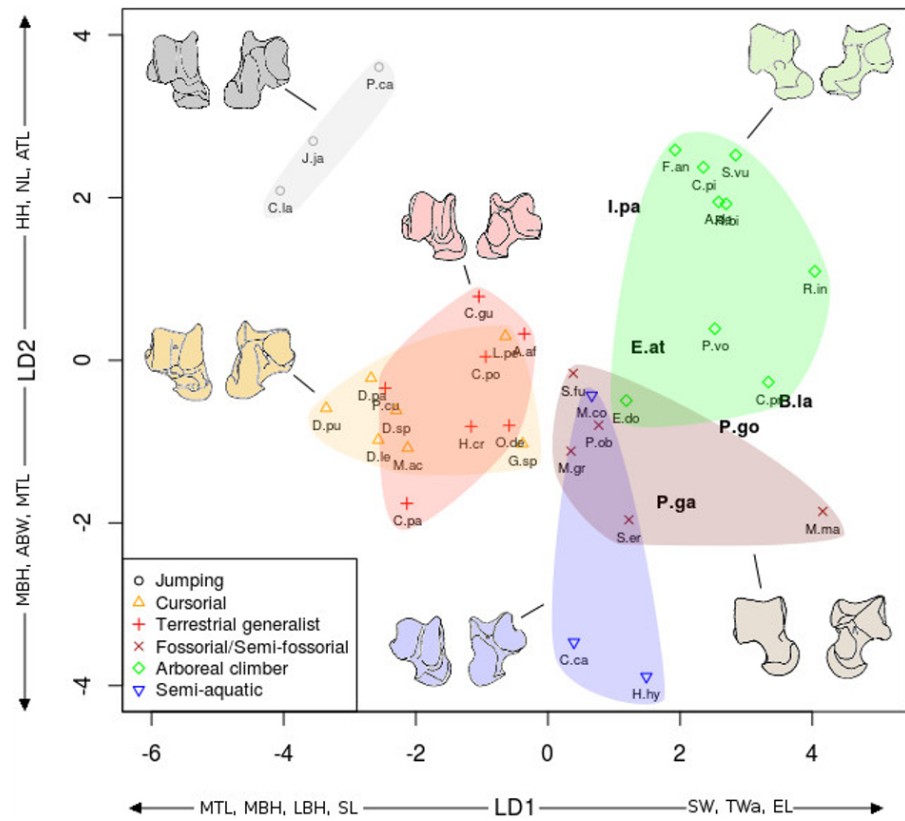

**Figure 6** **LDA of locomotory groups for the astragalus.** Linear discriminant analysis (LD1 and LD2) of locomotory groups based on mean log-shape ratios of linear measurements of the astragalus (see Fig. 2). Each point represents the average for a species (associated with abbreviated taxon name). In grey are the jumping taxa (locomotor category 1); in yellow are the cursorial taxa (category 2); in red are generalist taxa (category 3); in brown are the fossorial taxa (category 4); in green are the arboreal climbers (category 5); in blue are the semi-aquatic taxa (category 6). Drawings are here to show one morpological example of the corresponding category. Abbreviations in bold black font are the *a posteriori* placements of the fossil taxa: **B.la**, *Blainvillimys langei*; **E.at**, *Eucricetodon atavus*; **I.pa**, *Issiodoromys pauffiensis*; **P.go**, *Palaeosciurus goti*; **P.ga**, *Pseudoltinomys gaillardi*. Along axes are the main variables correlated with the linear discriminant functions (variables with an asterisk are close to significance). ABW, astragalus body width; ATL, astragalus total length; EL, ectal facet length; HH, head height; LBH, lateral body height; MBH, medial body height; MTL, medial trochlear length; NL, neck length; SL, sustentacular facet length; SW, sustentacular facet width; TWa, trochlear width.

taxa are found between cursorial and generalist taxa in the calcaneus LDA (Fig. 7), but closer to the fossorial/semi-fossorial group in the astragalus LDA (Fig. 6). Due to important differences between astragalus (Fig. 6) and calcaneus (Fig. 7), we treated the two bones separately in the following section.

The LDA allows for easier comparisons of log shape ratios between groups, both visually (Figs. 6 and 7) and by directly calculating the mean of each log shape ratio for each group. These means are given in Appendix 2 for the astragalus and Appendix 3 for the calcaneus. In the following section, group means are compared for variables which are significantly correlated with the linear discriminant functions (Table 2). In both the calcaneus and astragalus, MANOVAs showed locomotion and phylogeny had significant effects on the

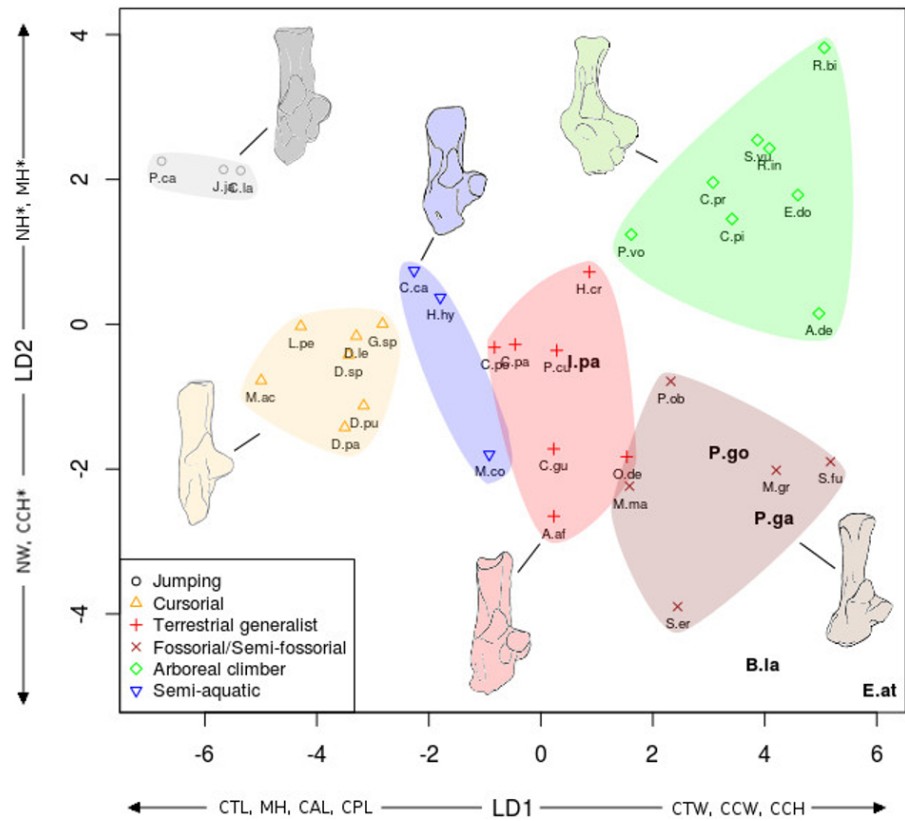

**Figure 7   LDA of locomotor groups for the calcaneus.** As Fig. 7 for calcaneus measurements. Abbreviations in bold black font are the *a posteriori* placements of the fossil taxa : **B.la**, *Blainvillimys langei*; **E.at**, *Eucricetodon atavus*; **I.pa**, *Issiodoromys pauffiensis*; **P.go**, *Palaeosciurus goti*; **P.ga**, *Pseudoltinomys gaillardi*. Along axes are the main variables correlated with the linear discriminant functions (variables with an asterisk are close to significance). CAL, calcaneus anterior length; CCH, calcaneo-cuboid facet height; CCW, calcaneo-cuboid facet width; CPL, calcaneus posterior length; CTL, calcaneus total length; CTW, calcaneus total width; MH, maximum height; NH, neck height; NW, neck width.

morphology of the bones (Astragalus locomotor category effect: Wilks' lambda = 0.167; $F = 5.3414$; $Pr(>F) < 0.002$; $Df = 1$; Astragalus phylogeny effect: Wilks' lambda = 0.0590; $F = 3.339$; $Pr(>F) < 0.001$; $Df = 2$. Calcaneus locomotor category effect: Wilks' lambda = 0.2728; $F = 2.6661$; $Pr(>F) < 0.05$; $Df = 1$; Calcaneus phylogeny effect: Wilks' lambda = 0.0649; $F = 2.9246$; $Pr(>F) < 0.003$; $Df = 2$). No significant effects of interactions between phylogeny and locomotion were found. The predictive power of each LDA was tested by leave-one-out cross validation. Both analyses placed about 40% of the taxa in the right locomotor category (41% for the astragalus and 36% for the calcaneus) for the extant taxa. It is should be noted that arboreal climbers and cursorial taxa get somewhat better scores with 67% (astragalus) and 50% (calcaneus) of right assignations for the climbers, and 43% (astragalus) and 57% (calcaneus) for the cursorial. Despite their isolated position in Figs. 6 and 7, no jumping taxa was placed correctly after cross-validation. Fossorial/semi-fossorial taxa, terrestrial generalists and semi-aquatic taxa have less than 40% correct assignations.

**Astragalus** (Fig. 6)

The first two LD axes (shown in Fig. 6) represent 73% of the variance (49% and 24% respectively). Variables correlated with the linear discriminant functions are shown along the axes in Fig. 6, and the correlation coefficients are presented in Table 2.

On the first axis, the jumping taxa are discriminated by high MBH and LBH, high SL and low SW values and low Twa, placing them along the negative part of the LD1. Cursorial and generalist taxa occupy the same morphospace, but are separated from semi-aquatic, fossorial/semi-fossorial, and climber taxa. This is linked to higher MBH, LBH and SL values and lower SW and EL values in cursorial and generalist taxa.

The second axis discriminates the semi-aquatic taxa on the negative side of the axis because of high ABW value, low ATL and HH values. Jumping taxa are also discriminated at the opposite side of the LD2, due to low ABW value, high ATL, HH and NL values. Climbers are also on the positive part of the second axis due to their high NL, low ABW, MBH and MTL values.

**Calcaneus** (Fig. 7)

The discriminant analysis performed on calcaneal measurements is different from the previous results based on the astragalar variables. With the calcaneus, the LDA does well at separating jumping and climbing taxa from all other categories. The first two discriminant axes (Fig. 7) show 89% of the variance (74 and 15% respectively). Variables correlated with the linear discriminant functions are shown along the axes in Fig. 7, and the correlation coefficients are presented in Table 2.

Most groups are separated along the first LD axis, with jumping and cursorial in the negative values on the axis, semi-aquatic and generalist taxa along medium values, and the climber and semi-fossorial/fossorial taxa in the positive values.

High CTL, CAL and CPL values explain the position of cursorial and jumping taxa. Jumpers have the highest MH value, placing them on the most negative part of the axis. Cusorial and jumping taxa also show low CTW and CCW values. Climbers and fossorial taxa are on the opposite side, characterized by high CTW, CCW and CCH values, combined with low MH, CAL, and CPL values. Semi-aquatic and generalist taxa have medium values, with the semi-aquatic group being more on the negative side due to higher CPL, and lower CTW values while the generalists are pulled to the positive side of the axis by a lower CPL and higher CCW and CTW values.

On the second axis the jumping and climbing taxa are discriminated from the rest, due to low NW values for the climbers and high MH for the jumpers.

## A posteriori fossil placement (Figs. 6–8)

The previous analyses allowed us to define the locomotory categories of fossil taxa. *Blainvillimys langei* (Theridomyidae) (Fig. 8A) is predicted to be a fossorial species in both the astragalus (posterior probability: 0.99) and calcaneus (0.99) analyses. The position of *B. langei* in Fig. 6 is closer to arboreal climber taxa, however Fig. 6 only represents LD1 and

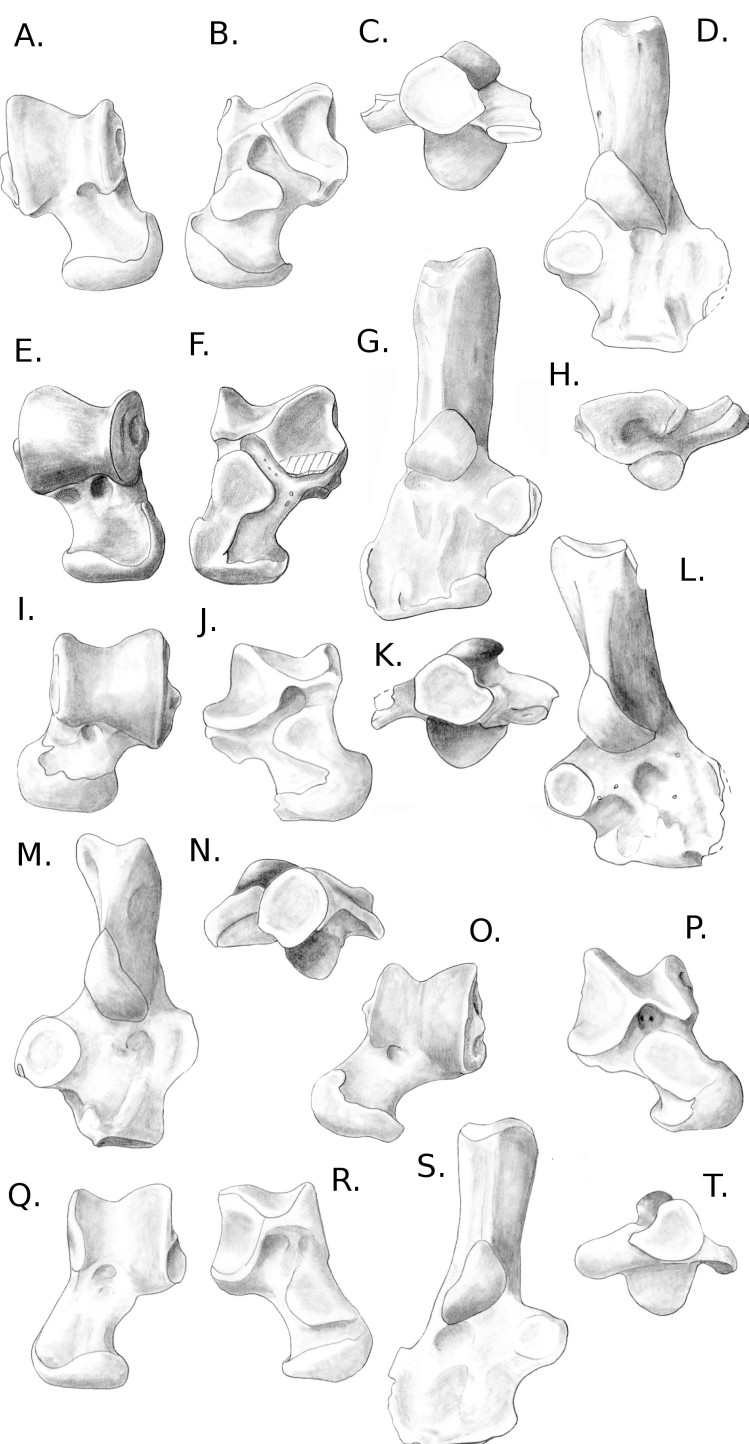

**Figure 8   Drawings of the calcanei and astragali of the fossil species studied.** (A) *Blainvillimys langei* RAV2001 (astragalus) and RAV2002 (calcaneus) (A1) astragalus, dorsal view, (A2) astragalus, plantar view, (A3) calcaneus, anterior view, (A4) calcaneus, dorsal view ; (B) *Issiodoromys Pauffiensis* SPV593 (astragalus) 

**Figure 8 (…continued)**
(B1) astragalus, dorsal view, (B2) astragalus, plantar view, (B3) calcaneus, dorsal view (specimen MPF213), (B4) calcaneus, anterior view (specimen SPV592); (C) *Pseudoltinomys gaillardi* RAV2003 (astragalus) and RAV2004 (calcaneus) (C1) astragalus, dorsal view, (C2) astragalus, plantar view, (C3) calcaneus, anterior view, (C4) calcaneus dorsal view; (D) *Palaeosciurus goti* MGB101 (astragalus) and MGB102 (calcaneus) (D1) calcaneus, dorsal view, (D2) calcaneus, anterior view, (D3) astragalus, dorsal view, (D4) astragalus plantar view; (E) *Eucricetodon atavus* RAV2005 (astragalus) and RAV2006 (calcaneus) (E1) astragalus, dorsal view, (E2) astragalus, plantar view, (E3) calcaneus, dorsal view, (E4) calcaneus, anterior view.

2, whereas the *a posteriori* placement is based on all linear discriminant functions taken together.

*Issiodoromys pauffiensis* (Theridomyidae) (Fig. 8B) is found to be a climber in the astragalus analysis (0.99) and a terrestrial generalist in the calcaneus (0.99) analysis.

*Pseudoltinomys gaillardi* (Theridomyidae) (Fig. 8C) is found as a fossorial (astragalus posterior probability: 0.99, calcaneus, 0.99) taxon.

*Palaeosciurus goti* (Sciuridae) (Fig. 8D) is predicted to be a fossorial species (astragalus posterior probability: 0.99, calcaneus : 0.96).

*Eucricetodon atavus* (Cricetidae) (Fig. 8E) is, inferred to be a fossorial species from both astragalus and calcaneus analyses (posterior probability: 0.99 and 0.99 respectively). Like *B. langei*, *E. atavus* is positioned within the climbers group in Fig. 6, again, the a posteriori inference of the locomotory group is different due to the influence of other linear discriminant functions.

Both bones give fairly similar predictions, with some discrepancies in *I. pauffiensis*. The apparent differences between these predictions and the results of the LDA analyses (Figs. 6 and 7) are due to the fact that the predictions are based on all linear discriminant axes combined, while Figs. 6 and 7 only show LD1 and LD2. The high posterior probabilities found for these fossil species should be taken carefully, as only around 40% of placements are correct for extant taxa.

## DISCUSSION

### Functional analysis of locomotor types

The discriminant variables found in the LDA can be combined with the qualitative characters of the osteological descriptions. In this part we describe the position of the groups, based on the quantitative variables, then combine it with qualitative data, to detect how morphologies converge within locomotor groups.

### Quantitative differences between locomotor groups
#### Astragalus

The astragalus body width (ABW) is greatest in semi-aquatic taxa due to the lateral prominence of the ectal facet. It is also high in generalist taxa for the same reason. The high value observed in fossorial/semi-fossorial taxa is linked to the fairly wide trochlea itself, associated with a slight lateral projection of the sustentacular facet. Climbing taxa have a low mean value because their sustentacular facet is not projected laterally. In jumping taxa, the trochlea is compressed transversally explaining the low mean value.

The astragalus total length (ATL) is much higher in jumping taxa than in any other category. This is due to an antero-posterior lengthening as well as a transversal compression of all structures of the bone. The lowest value is observed in semi-aquatic taxa, due to the robust morphology of the astragalus.

The astragalus total width (ATW) is highest in generalists due to their wide trochlea, the lateral projection of their ectal facet, as well as a slightly deviated neck. Climbers and fossorial/semi-fossorial taxa also have a fairly high value linked to the medial projection of the neck. The lowest values are found in jumpers, cursors and semi-aquatic taxa, in which the neck is not as much laterally projected. Furthermore, jumping taxa exhibit a transversal compression of the bone.

The ectal facet length (EL) shows only slight differences. The highest values are found in climbers, fossorial/semi-fossorial, and generalist taxa. In these taxa, the ectal facet is not projected laterally, but extends medially and antero-posteriorly. Furthermore, it is less curved (and therefore longer), which gives flexibility to the ankle. Jumping and cursorial taxa display the lowest values due to the strong curvature of the facet, limiting the movements of the calcaneus in regard with the astragalus.

The head height (HH) is low in cursorial and semi-aquatic taxa, mainly due to the overall reduction in size of the head. Functionally, this could imply less flexibility. Highest values are found in jumping taxa in which the shape of the head allows for a wide array of parasagittal movements of the foot.

The head width (HW) is quite similar in all categories. Fossorial/semi-fossorial and cursorial taxa have the widest astragalar head, enabling some flexibility to the foot independently of the ankle, since the head of the astragalus articulates anteriorly with the navicular. At the other end, semi-aquatic taxa have the narrowest head, greatly limiting transversal movements of the foot independently of the ankle.

The lateral and medial body heights (LBH and MBH respectively) are lower for climbing and fossorial taxa, due to a flatter shape of the trochlea in medial and lateral views. This trochlear shape is, in turn, linked to a lesser arc of parasagittal movements in these taxa. Conversely, jumping and cursorial taxa have highest lateral and medial body height, due to a greater curvature of the trochlea, which relates to a wide arc of parasagittal movement.

The lateral trochlear length (LTL) is very similar in all categories, except for the semi-aquatic taxa, for which it is shorter. Again, this is due to a short and more symmetrical astragalus in these taxa. The medial trochlear length (MTL) is smallest in climbers and fossorial/semi-fossorial taxa, where a marked asymmetry is noticeable between the lateral and medial parts of the trochlea (i.e., LTL and MTL are markedly different). This asymmetry adds a transverse component to the plantar and dorsiflexion of the foot. Jumping and generalist taxa also show a medium asymmetry, while cursorial and semi-aquatic taxa have an almost perfect symmetry of the trochlea. The reduction of the asymmetry means the foot will stay on the parasagittal plane during plantar or dorsiflexion of the foot, allowing for more efficient ruuning/paddling.

The neck length (NL) is highest in jumping taxa, in which the neck is lengthened antero-posteriorly. The neck is also long in fossorial and arboreal climber taxa, in part because it is deviated medially. This gives a greater flexibility to the foot, either in the

parasagittal plane (for jumping taxa) or medially (for fossorial/climber taxa). Other taxa have a shortened neck, suggesting less flexibility of the foot in regard with the ankle.

The sustentacular length (SL) is higher in jumping, cursorial and semi-aquatic taxa, where it favours parasagittal movements. In fossorial taxa, the antero-posterior dimension of the sustentacular facet is generally reduced, limiting antero-posterior mobility of the astragalus in regard to the calcaneus. Indeed, in fossorial taxa, these bones should be maintained together during parasagittal movements (e.g., pushing back the soil).

The sustentacular width (SW) is highest in fossorial and climber taxa. The transversely oriented shape of the sustentacular facet permits medio-lateral movements, crucial for these locomotor categories. In jumping and cursorial taxa, the facet is transversely compressed (i.e., SW is small), again in accordance with the nature of the movements of the foot, which are mainly parasagittal.

Trochlear width (TWa) is almost similar in all categories. Jumping taxa have the lowest values and show a transversely compressed astragalus , while arboreal climbers show a high value and are characterized by asymmetrical and slightly divergent medial and lateral parts of the trochlea. The widest trochlea is found in semi-aquatic taxa, in which the astragalus is shortened and widened. Fossorial/semi-fossorial taxa also have a fairly wide trochlea, bringing some stability to the upper ankle joint.

### Calcaneus

The calcaneus anterior length (CAL) is highest in jumping and cursorial, it is intermediate in generalist taxa, and the lowest values are found in fossorial, arboreal climbers and semi-aquatic taxa. Morphologically, this means that the body of the calcaneus, situated anteriorly to the ectal facet, is lengthened in terrestrial taxa, especially specialized cursorial and jumpers taxa, while it is shortened in taxa specialized in other substrates.

The posterior length (CPL) is highest in cursorial and semi-aquatic taxa. It is also high in jumping taxa, while generalists, as well as climbers, show lower values. The lowest mean value is found in fossorial/semi-fossorial species.

The calcaneo-cuboid height (CCH) and the calcaneo-cuboid width (CCW) define the shape of the calcaneo-cuboid facet. In general, its shape differs in anterior view. This facet is biggest (i.e., CCH and CCW have the highest values) in fossorial/semi-fossorial and arboreal climber taxa. In these species, it has an elliptical shape, whereas it is more slender in other categories, with a dorso-plantar main axis, and a transverse compression. This elliptical shape of the facet allows a variety of movements of the foot, through the cuboid, in fossorial/semi-fossorial and climber taxa. In other taxa, transverse movements of the foot are generally more limited, due to a narrower facet (i.e., CCW has lower values in terrestrial and semi-aquatic taxa).

The calcaneal ectal length (EL) measurements can differ from those of the ectal facet of the astragalus. The jumping and semi-aquatic taxa display low values (same as the astragalus) due to the much curved shape, and the perpendicular orientation of the facet in dorsal view. Conversely, in climbers and generalists taxa, the calcaneal ectal facet is less curved, mainly oriented antero-posteriorly, and posteriorly lengthened to increase mobility

between the two bones. Fossorial taxa show a low value, whereas they showed a high value for the astragalar ectal facet.

EW is highest in cursorial and semi-aquatic taxa, and lowest in jumping, fossorial/semi-fossorial and generalist taxa. Climbers show intermediate mean value. This variation does not match that of the astragalar ectal facet. On the calcaneus, EW is only correlated with LD4 and LD5, and on the astragalus it is not correlated with any linear discriminant axis. It seems that the EW does not display a strong ecomoprhological signal.

The maximal height (MH) of the calcaneus is highest in jumping taxa. It is also intermediate in cursorial, generalist and semi-aquatic taxa. Lower values are found in fossorial/semi-fossorial and climbers taxa. The variation in MH is linked mainly to the orientation and projection of the ectal facet (see Fig. 2). The high values found in jumping, cursorial, generalist and semi-aquatic taxa are therefore linked to their ectal facet being oriented perpendicularly to the antero-posterior axis of the bone.

The neck width (NW) is lowest in climbers taxa. This is linked to the curvature of the neck and the "pinch" that is visible at the middle of the neck in these species (e.g. Fig. 3D). Cursorial taxa show an intermediate value, while other categories have higher value, explained by a more robust neck shape.

The calcaneal sustentacular length (SL) is reduced in cursorial, fossorial and generalist taxa, where the shape of the facet does not facilitate antero-posterior movements of the calcaneus with regard to the astragalus. Conversely, the SL in jumping, semi-aquatic, but also climbing taxa is high, meaning that the bones have a relative independence in antero-posterior movements in these taxa.

The sustentacular width (SW) is low in taxa that mainly use parasagittal movements (i.e., mainly jumping and cursorial but also semi-aquatic). Fossorial, climbers and generalists taxa, in which the ankle is more mobile transversely, show higher values.

The tuber height (TH) is greatest in fossorial and climber taxa. Generalists display an intermediate value, while jumping, cursorial and semi-aquatic taxa have lower average TH. The shape of this area is linked with the insertion of the Achilles tendon.

Jumping and fossorial/semi-fossorial taxa have the highest TWc values, while generalists, semi-aquatic and cursorial taxa have intermediate values. All these taxa show a rather robust tuber morphology. Conversely, arboreal climbers have a very low TW value, linked to a more slender tuber.

## Morphology and function in the locomotor groups
### Jumping and cursorial taxa (Fig. 9)

Jumping and cursorial taxa are treated altogether because their calcanei and astragali show morphological similarities and have apparently similar functional constraints (rapid parasgittal movements). However, they also show differences, which are clearly underlined in the discriminant analyses and will also be surveyed.

**Astragalus**. In these taxa, the trochlea of the astragalus is almost symmetrical, with rims aligned in the parasagittal plane. The trochlea is also transversely compressed (low TW value, especially in jumping taxa). In this context, the foot moves in the parasagittal plane when rotated (*Carrano, 1997*; *Candela & Picasso, 2008*). The trochlear groove is deeper,

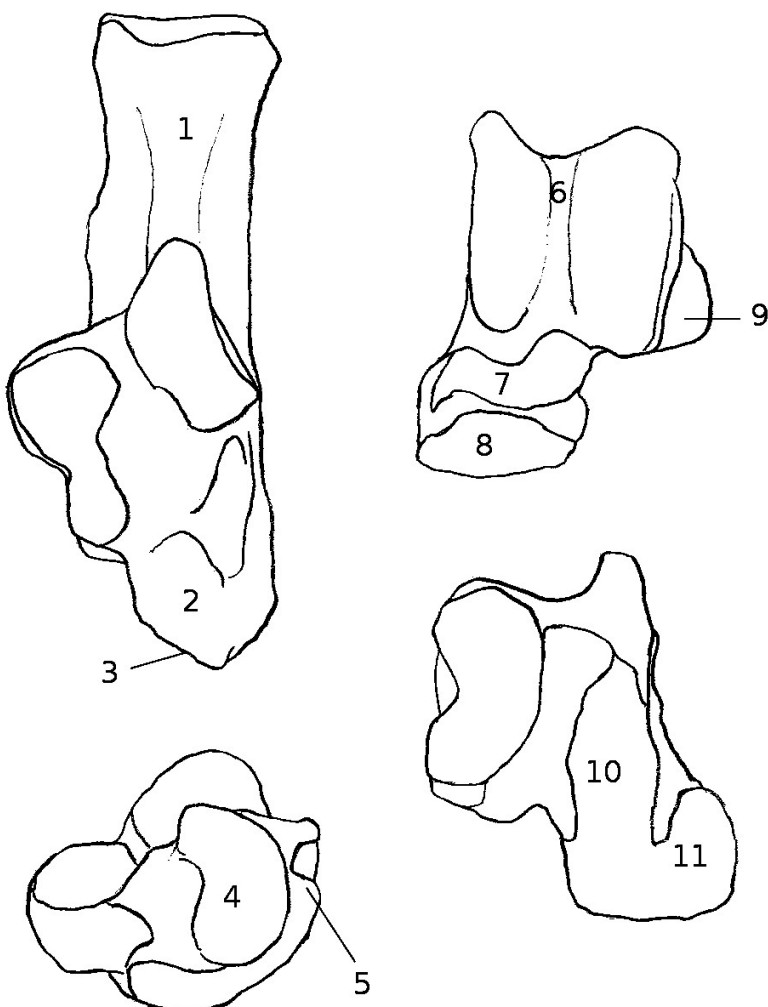

**Figure 9 Drawing of the calcaneus and astragalus of a "typical" jumping rodent.** Calcaneus and astragalus of *Chinchilla lanigera* illustrating jumping morphology. Numbers represent typical qualitative characters for this locomotory group, as well as most of the cursorial taxa studied here. (1) Neck (calcaneus) straight, and (2) of equivalent width with the anterior part of the bone. CaCu facet (3) oblique and (4) bean or crescent-shaped. (5) Peroneal process not developed and in distal or distal-most position. (6) Symetrical trochlea with a deep groove. (7) Neck (astragalus) slightly or not deflected medially. (8) Head narrow, with the AN facet developed dorsally. (9) Ectal facet projected laterally. (10) Sustentacular facet slender and antero-posteriorly oriented. (11) AmT facet developed on the plantar and/or medial side of the neck (astragalus).

and both sides of the trochlea are steeper than in other taxa. This restricts or may even suppress any transverse movement when the foot is dorsally or plantarly flexed (*Carrano, 1997*; *Candela & Picasso, 2008*). In these taxa, limited transverse movements may be of major importance to avoid injuries at the ankle (*Hildebrand, 1985a*; *Hildebrand & Goslow Jr, 2001*). The astragalar neck is of variable lengths, longer in jumping taxa (high NL value), in which it participates to the great lengthening of the foot seen for example in gerboas. It is shorter in cursors, which have a low average NL value. In both groups the neck is barely or not deflected medially. As a consequence, the sustentacular facet (of

variable length, depending on the neck) is always anteroposteriorly oriented. This greatly reduces the transverse movements of the calcaneus below the astragalus (*Candela & Picasso, 2008*), making the inversion of the foot impossible or extremely difficult. In addition, the sustentacular and ectal facets are usually separated by a deep and/or wide sulcus, which suggests the presence of strong ligaments between the two bones (Fig. 1B) that make them less independent in their respective motions (conversely to what is seen in flying squirrels, see *P. volans*; *Thorington Jr et al., 2005*). The deep sulcus and antero-posterior sustentacular facet are, again, diplays of the great resistance of the ankle necessary in these two types of locomotion (*Hildebrand, 1985a*; *Hildebrand & Goslow Jr, 2001*). The ectal facet of the astragalus is generally strongly projected laterally, and its calcaneal counterpart matches perfectly with it. This limits the transverse and anteroposterior mobilities of one bone with regard to the other. In these taxa, the astragalus and calcaneus are more tightly linked than in climbers, moving as a single unit rather than as independent entities. The distal articulation with the navicular is different in cursorial and jumping taxa. The former have a very low HH value and rather high HW, while the latter have high HH and intermediate HW. In functional terms, this suggests that the anterior part of the foot in jumping taxa has more mobility in regard with the ankle than in cursorial taxa. This may allow jumping taxa to gain some elastic force when landing, and/or to cope with stresses produced on the lengthened metatarsals (injury avoidance; *Hildebrand, 1985a*; *Hildebrand & Goslow Jr, 2001*). One final characteristic element is the AmT facet, which can be of variable length, but is barely or not visible on the neck in dorsal view. It is actually developed on the medial and plantar sides of the neck. Therefore, the medial tarsal is positioned plantar to the astragalus, providing resistance against the reaction forces (directed upwards) transmitted to the ankle during locomotion (*Hildebrand, 1985a*; *Hildebrand & Goslow Jr, 2001*).

**Calcaneus.** The calcaneal ectal facet is usually shorter in jumping than in cursorial taxa (lowest EL value for jumping taxa)but its anterior part is always oriented perpendicularly to the dorsal surface of the body of the bone and corresponds to the strong curvature of the ectal facet of the astragalus (low astragalar EL in both groups). As a whole, the shape of the calcaneus is straight and aligned in the parasagittal plane. The neck is straight and of regular width. The body is usually long and rather narrow (lowest CTW values in jumping and cursorial taxa) in comparison with other taxa. This shape shows that transverse movements are probably very infrequent and favors parasagittal motions instead. The calcaneo-cuboid facet is oblique in dorsal view, crescent or bean-shaped, and a dorso-plantar main axis. Such a shape implies again that the cuboid has limited transverse movements with respect to the calcaneus (*Candela & Picasso, 2008*). The morphology of the facet seems to be convergent with other cursorial mammals such as some artiodactyls (e.g., Bovinae and Caprinae, pers. obs.). The peroneal process is reduced or absent, therefore changing the lever system of the *peroneus* muscle group into a direct pulling (with no fulcrum, see Fig. 1 and Table S1). This should greatly reduce the possibility of abducting and/or everting the foot, contrary to climber taxa.

*Terrestrial generalists*

**Calcaneus**. Overall, it has a robust shape, not displaying marked lengthening (intermediate CAL and CPL values). The peroneal process is sligthly developed laterally (contrary to cursors/jumpers where it is absent), and in an anterior position (unlike climbers/fossorial taxa where it is more posterior). One exception is *C. porcellus* (Fig. 5K) where the process is absent, similar to cursorial and jumping taxa. Ectal and sustentacular facets are not much developed (low average SL and EW and intermediate EL values). Both facets are always separated by a sulcus, except for *Ctenodactylus vali* (Fig. 5J) where they form a continuous articular surface. The calcaneo-cuboid facet is developed and with a wide crescent shape, with a CCW average value intermediate between cursorial/jumping taxa and fossorial/climber taxa. In *C. porcelus*, it is smaller than in other generalists (Fig. 5K). The neck is straight and robust in all generalist taxa.

**Astragalus**. The ectal facet is not projected laterally (except in *O. degu* and *P. cuvieri*, Figs. 5N and 5P, respecctively). Ectal and sustentacular facets are separated by a sulcus, except in *C. vali*, corresponding to what is seen in the calcaneus (Figs. 1A and 1B). The head is projected laterally and developed with a rather round shape, except in *C. porcellus*, where it is less developed in width and length (Fig. 5K). The astragalo-navicular facet is developed on the dorsal side of the neck, as well as AmT facet. The sustentacular facet has an ovoid shape (intermediate SL and SW values) and does not join with the AN facet, except in *C. porcellus* where it is extended and joins anteriorly with the AN facet. The trochlea is rather symmetrical (high LTL and MTL average values), with an deep groove . It appears to be more asymetrical in *O. degu* than in other generalist taxa.

Overall the generalists display characters suggesting a rather good flexibility at the ankle. It should be noted that *Cavia porcellus* shows some cursorial characterisitcs (lengthened astragalar sustentacular facet, absence of peroneal process), and that *Octodon degu* seems to have more flexibility than other, with some characters similar to climbers (continuous ectal and sustentacular facets, asymmetrical trochlea). It is also important to note that all generalist taxa studied here are members of the Ctenohystrica, which may also influence the similarities in morphology. However, members of the Ctenohystrica which are not generalists display different morphologies, suggesting that the qualitative characters described before are not influenced solely by phylogenetic proximity.

*Arboreal climbers (Fig. 10)*

**Calcaneus**. The calcaneus of climber taxa shows common features. The medial curvature of the neck may be linked to the direction of the forces that are strongest and most often exerted (i.e., during inversion and plantar flexion of the foot) by the muscles on the tuber (*gastrocnemius* and *soleus*, see Table S1). Indeed, each time the calcaneus pivots medially, its medial side anatomically moves to a more dorsal position. Therefore, the muscle stresses are not exerted on the dorsal side of the neck but on its medial side, which is in a dorsal position (Table S1 and below). During this movement, the bending of the neck of the calcaneus may provide a stronger resistance to the large forces applied, or may be due to the fact these stresses were present during the growth of the bone and influenced its shaping. Interestingly, some arboreal marsupials display a similar character state (such as *Caluromys philander* in:

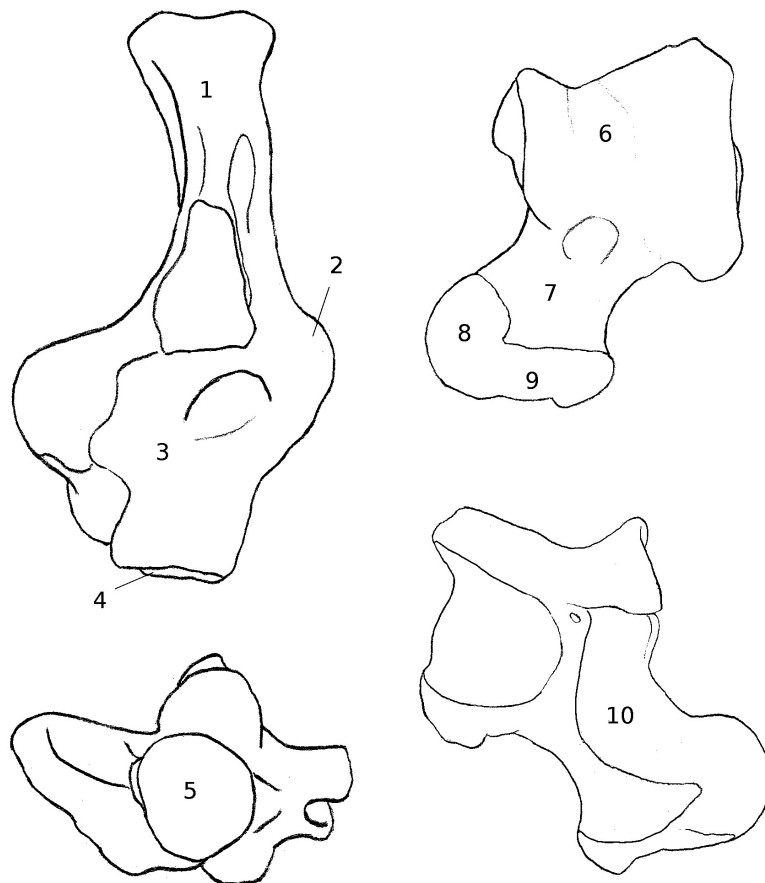

**Figure 10** **Drawing of the calcaneus and astragalus of a "typical" climbing rodent.** Calcaneus and astragalus of *Sciurus vulgaris* illustrating arboreal climber morphology. Numbers represent typical qualitative characters for this locomotory group. (1) Slender and medially curved neck (calcaneus). (2) Peroneal process well-developed and in a posterior position. (3) Wide anterior part of the bone. Calcaneo-cuboid facet (4) perpendicular to the main axis of the bone and (5) of circular shape. (6) Asymetrical trochlea with a shallow groove. (7) Medially deflected neck (astragalus). (8) AmT facet developed on the dorsal side of the neck. (9) Wide and round head. (10) Sustentacular facet wide and oriented transversally.

*Szalay, 2006*, p. 193, Fig. 7.8). The importance of the stress produced by the *gastrocnemius* muscle may explain the high value of trochlear height (TH) found in climbers. Indeed, the TH is influenced by the development of the Achilles tendon insertion area (Fig. 1). In climbers, this insertion is probably strenghtened in order to sustain the stress induced during inversion. The peroneal process is well-developed in all climber taxa studied; it is also in a posterior position on the lateral side of the bone (*Emry & Thorington, 1982*; *Thorington Jr et al., 2005*, Fig. 10; *Rose & Chinnery, 2004*; *Candela & Picasso, 2008*). The process forms a fulcrum for the tendons of *M. peroneus longus* and *brevis*, abductors/eversors of the foot (Fig. 1; Table S1). A wide and posterior peroneal process may therefore improve the mobility of the foot by facilitating its eversion and/or abduction through a change in the characteristics of the lever system of the *peroneus* muscles. In that case, the lever system is a "type 3 lever," where the effort is exerted between the fulcrum and the resistance (*Davidovits, 2012*). Therefore, the posterior position of the peroneal process increases the

in-lever length, producing a greater mechanical advantage for the *peroneus* muscles. It should also increase the resistance to forces on the opposite (medial) side during inversion. The shape and orientation of the calcaneo-cuboid facet (slightly concave, circular, and oriented transversely or only slightly oblique) also allows for a great variety of movements at the level of the cuboid bone (*Candela & Picasso, 2008*). The circular shape of this facet in climbing taxa explains the high CCW and CCH values found in that group. This last character state, associated with the shape of the talar head, gives a great deal of flexibility to the transverse tarsal articulation, and consequently makes the distal part of the foot more independent from the ankle (*Candela & Picasso, 2008*). In climbers, this flexibility is probably needed to cope with the varying slope and irregularities of the substrate.

**Astragalus**. The trochlea of the astragalus of frequent climbers is generally very asymmetrical. As such, the rotation axis of the foot is not parasagittal but has a transverse component (*Szalay, 1985*; *Candela & Picasso, 2008*). The foot can therefore adapt to substrates of variable inclination, usually by different degrees of inversion/eversion positions (*Szalay, 1985*; *Carrano, 1997*; *Candela & Picasso, 2008*; *Schmidt & Fischer, 2011*). Furthermore, the trochlear groove is shallow, and both the lateral and medial parts are generally not steep towards it. *P. volans* shows a sharper groove than *S. vulgaris*. This character was described in flying squirrels as a more "V"-shaped trochlea by *Thorington Jr et al. (2005)* (however, we would describe it as a more "checkmark"-shaped trochlea, considering the asymmetry of the lateral and medial sides). This allows the upper ankle joint (UAJ) to have some lateral mobility, even when the foot is dorsally or plantarly flexed (*Szalay, 1985*). The neck of the astragalus may be long, but it seems that a strong medial deviation might best characterize climbing rodents. Indeed, all climber taxa studied qualitatively here (*Sciurus*, *Pteromys*, and *Coendou*) have a medially deflected neck, but only the squirrels have a particularly long neck. An elongated and/or deflected neck increases the articular surface of the plantar sustentacular facet, along which the calcaneus may slide. Furthermore, the orientation of the facet, following the axis of the neck, favours medial movements, which are particularly required to invert the foot position (*Szalay, 1985*; *Candela & Picasso, 2008*). In relation with the previous points, the broadening of the ectal and sustentacular facets, or their confluence, creates a more continuous surface of articulation and reduces the attachment of ligaments (notably with the disappearance of the sulcus in *P. volans*). The absence of a sulcus between the facets in flying squirrels has also been noted by *Thorington Jr et al. (2005)*, who linked it to the extremely inverted position of the foot during gliding. In any case, the combination of morphological features found in all climbing rodents studied here facilitates any potential movements of the calcaneus along the articular surfaces of the astragalus. This, in turn, allows the foot to move more independently with respect to the leg, thanks to a flexible ankle (*Szalay, 1985*; *Candela & Picasso, 2008*). The AmT facet is developed on the dorsal and medial aspects of the astragalar neck. Therefore, when important constraints are applied to the medial side of the astragalus (i.e., during inversion of the foot while climbing), the medial tarsal bone can support the astragalus by taking a more medial or dorsal position, thereby stabilizing the ankle (*Emry & Thorington, 1982*; *Szalay, 1985*; *Candela & Picasso, 2008*). The proximo-plantar edge of the trochlea ends in a salient medial tuberosity, which will stop the medial side of the neck of the

calcaneus in a maximal foot inversion (although separated by different layers of non-osseous tissue). This character also maintains and stabilizes the ankle joint during inversion of the foot. Finally, the wide, sometimes rounded astragalar head, together with the confluence of its articular facets, show that the distal tarsal bones articulating with it (navicular and medial tarsal) may take various positions with regard to the astragalus. This mobility of the distal tarsals translates in a great flexibility of the distal part of the foot, again required by the use of irregular and inclined substrates (*Szalay, 1985*; *Candela & Picasso, 2008*).

## Fossorial taxa

The semi-fossorial ground squirrels (*Marmota* and *Spermophilus*) show an interesting combination of traits similar to arboreal sciurids (e.g., a strong peroneal process, a long and deflected astragalar neck, an asymmetrical trochlea), although with some notable differences: the trochlea is not as asymmetrical as in arboreal climbing squirrels, and the peroneal process, although well developed is in a more anterior position (*Emry & Thorington, 1982*). These characters could be ancestral to the group since some of the earliest sciuroids were probable climbers (e.g., *Douglasciurus* Douglas, 1901; *Emry & Thorington, 1982*; *Vianey-Liaud, Gomes-Rodrigues & Marivaux, 2013*). The more anterior position of the peroneal process, the deep sulcus between the ectal and sustentacular facets, the astragalar sustentacular facet less medially developed, or the short calcaneal body in comparison to the calcaneal neck, characterize terrestrial rather than arboreal forms (*Szalay, 1985*; *Carrano, 1997*; *Candela & Picasso, 2008*; *Thorington Jr et al., 2005*).

In *Spalax* (the only fully subterranean species studied here), the calcaneo-astragalar complex combines a good mobility at the ankle, characterized by a wide trochlea with a shallow groove, long astragalar neck with a spherical head, long calcaneal ectal facet, developed peroneal process; with a powerful plantar stroke, thanks to the elongation of the in-lever arm that produces a greater force during extension (e.g., elongated calcaneal neck with a deep and wide tuber, short calcaneal body; *Carrano, 1997*; *Davidovits, 2012*). This is likely to help maintain the animal while digging and/or to push the dirt out of the way (*Hildebrand, 1985b*).

The peroneal process is developed in *Spalax*, as well as in semi-fossorial species (*Marmota Marmota*, *Spermophilus fulvus*, and *Microtus gregalis*). Therefore, the eversion and abduction of the foot are probably important elements in fossorial behaviours, as well as in arboreal climbing, although they do not serve the same purpose. In semi-fossorial species, the neck of the calcaneus bears well-marked muscular insertion areas, especially in *Marmota*. Like in *Spalax*, this is certainly linked to the power of the lever arm of the foot, crucial in digging, even though *M. marmota*, *S. fulvus* and *M. gregalis* cannot be considered as complete fossorial species. The calcaneal ectal facet is not lengthened in fossorial taxa (low EL value), and neither is the sustentacular facet (low SL value for the astragalus and calcaneus). Indeed, the astragalus and calcaneus should be maintained together under important constraints, rather than be allowed to move independently. In fossorial and semi-fossorial taxa, the flexibility of the ankle mainly depends on the UAJ, as well as the astragalo-navicular joint (high HW value with a developed and convex AN facet).

It is interesting to note that the peroneal process length (PPL) is highest in fossorial species. The development of this process is linked to the passage of the tendons of the *peroneus* muscle group (Fig. 1), which are responsible for the eversion of the foot. The importance of the peroneal process in fossorial taxa may be explained by the movements used to push back the soil while digging (possibly a partially lateral movement, rather than just a backwards parasagittal movement). The peroneal process is also present and developed in climber taxa (second highest PPL average score), in which eversion is frequent during climbing. It appears the less a taxon is using transversal movements of the foot, the more reduced its peroneal process is (e.g., semi-aquatic and jumping taxa have the lowest PPL values).

### Semi-aquatic

Although the castor and coypu have different morphologies for their astragalus and calcaneus, probably due to their distant phylogenetic relationship, we can identify several convergent character states.

**Calcaneus**. The calcaneus of both species bears an ectal facet that is oriented towards the medial side. This may be linked to an increase in the parasagittal arc that the foot may produce when swimming, therefore improving the hindlimb propulsion. In both species, the sustentacular facet is concave, which may improve the stability of the astragalus on the calcaneus during movements. Finally, both species display a very deep and wide groove for the *flexor fiblaris* muscle (Fig. 1A). This muscle plays a role in parasagittal movements of the foot (Table S1), especially plantar flexion, which is important when paddling . The importance of the groove for the tendon of this muscle suggests that the muscle is very active in semi-aquatic species.

**Astragalus**. The astragalus facets appear very similar. Indeed, the ectal facet is wide (highest EW mean value), mirroring that of the calcaneus, and suggesting that the ankle may have some flexibility in semi-aquatic species. If fhe sustentacular facets have very different shapes, they are both mainly oriented antero-posteriorly (highest mean SL value), as in other taxa where dominant movements of the foot are parasagittal. Finally, the neck shows slight deviation towards the medial side in both species, suggesting again that there may be moderate flexibility in the ankle of semi-aquatic taxa.

### Proportions of the calcaneus as the lever arm of the foot

Since the calcaneus can be seen as a lever (*Carrano, 1997*; *Davidovits, 2012*), the functional significance of the CAL and CPL values appears more clearly through a ratio of in-lever/out-lever proportions. If $F$ is the force necessary to move the foot, CAL can be considered as the out-lever, CPL as the in-lever, and $k$ as the resistance force of the substrate (*Davidovits, 2012*). Therefore, we have $F = (CAL/CPL) + k$. Using logshape ratios values for CAL/CPL, the results are as follows : jumping, 1.06 ; cursorial, 0.69; generalist, 0.88 ; fossorial, 0.40 ; climber, 0.45; semi-aquatic, 0.24.

The ratio is lower in fossorial, climber and semi-aquatic taxa. Such ratios enable these taxa to compensate for greater resistance forces ($k$). For climbing taxa, $k$ may be increased by the effects of gravity. In fossorial taxa, pushing back the soil would be the main resistance force applying, however, the CAL/CPL value observed is not as low as in semi-aquatic

taxa. This may be explained by the fact that the hindlimbs are used to brace the animal (*Hildebrand, 1985b*) and push out the soil during digging, rather than to do digging itself. In semiaquatic taxa, water resistance could be one element to account for increasing $k$. However, the taxa studied here are quite often burrow diggers, and this behaviour could also partly explain the $k$ value observed here. Interestingly, the coypu does not group with the castor and capybara in Figs. 6 and 7. This may be due to the smaller size of the former, creating less drag when swimming, and therefore requiring less morphological specialization to overcome it.

## LDA interpretation caveats

The LDAs allow us to determine which variables show inter-group variance (i.e., between locomotor categories). Since these variables are significantly correlated with locomotory categories, they can be used to infer the locomotory group of a specimen with some certainty. However, the Wilks lambda values are rather high, and cross validations are not very successful, which suggest that the predictions should be taken with caution. Still, the morphology shows locomotory signal, and we can be fairly confident in our inferrences, notably for for the cursorial and climber taxa.

The differences in the group discriminations observed between astragalus and calcaneus can be explained functionally. Indeed, although these bones are part of the same complex unit (the ankle), they have quite different functions. The astragalus is a pivot, the *fulcrum* of the foot as a lever (*Carrano, 1997*; *Davidovits, 2012*); it must ease or limit movements, depending on the favoured mode of locomotion. On the other hand, the calcaneus is part of the active movement; it is the lever arm (*Carrano, 1997*; *Davidovits, 2012*) that transmits the energy from muscular effort. In other words the calcaneus applies forces, whereas the astragalus only guides the movements. For instance, the astragalus of fossorial rodents is show some similarities with that of arboreal climber taxa, probably because they both need to have ankles that show some sort of flexibility (for climbing in arboreal taxa, or for pushing soil and maneuvering in tunnels for fossorial taxa). We should therefore use caution when making interpretations based on an isolated bone.

## Inference of locomotion in fossil rodents (Figs. 6–8)

Discrepancies between inferences from astragalus and calcaneus are present in one case (*Issiodoromys pauffiensis*). It is one of the reasons why *a posteriori* placement in a LDA should be used with caution when inferring the locomotor group of a fossil species, especially when only one bone is available. Of course the low success of cross validation is the main problem with these inferrences. Still, discrepencies within taxa, between bones are not surprising and can be explained by the different roles that each bone of interest can play during movements (see preceding paragraph). Fossorial and generalist taxa may be further confounded by the fact that terrestrial generalists may also be burrow diggers (e.g., *Spermophilus fulvus*, *Tsytsulina, Formozov & Sheftel, 2008*; *Octodon degus*, *Woods & Boraker, 1975*).

Regarding *Palaeosciurus goti*, the result is in accordance with *Vianey-Liaud (1974)*, which described it as a "terrestrial squirrel," rather than an arboreal climber. The astragalus, when taken as the fulcrum of the foot-lever (*Carrano, 1997*; *Davidovits, 2012*), displays the flexibility required in the digging process (Figs. 8B3 and 8B4, right). The calcaneus
also matches with that of fossorial/semi-fossorial taxa. Since all terrestrial squirrels studied here are also semi-fossorial, it is fairly plausible that *P. goti* had a similar ecology and locomotion. *Vianey-Liaud, Hautier & Marivaux (2015)* defined the lifestyle of *P. goti* as scansorial/terrestrial (*Vianey-Liaud, Hautier & Marivaux, 2015* Supp. data, Tab. 10.6) and proposed that *P. goti* would have been an occasional climber. The result presented here does not exclude this possibility since some of the terrestrial generalists taxa studied are also occasional climbers (e.g., *Ctenodactylus vali*, Fig. 5J, *Gouat & Gouat, 1987*; *O. degus*, Fig. 5O, *Woods & Boraker, 1975*). Furthermore, the astragalus presents a morphology allowing flexible movements of the ankle, which may have been of use either for digging or climbing (climbers and diggers are close in Fig. 6 because of some similar functional constraints). However, although occasional climbing may have been possible in *P. goti*, the qualitative characters seem in accordance with a semi-fossorial lifestyle. When comparing with other fossil and extant squirrels (*Emry & Thorington, 1982*; *Korth & Samuels, 2015*), it appears that the calcaneus of *Palaeosciurus* has a shorter and wider neck than *Douglassciurus* (*Emry & Thorington, 1982*) or *Protosciurus* (*Korth & Samuels, 2015*). The neck is only slightly deflected medially, and the medial part of the tuber is more prominent than the lateral part, similar to *Protosciurus* (*Korth & Samuels, 2015*). The ectal facet has a shape closer to that of *Protosciurus* (described as "trapezoidal" in *Korth & Samuels, 2015*) than to *Douglassciurus* or *Sciurus*. The sustentacular facet is round and separated from the ectal facet by a deep groove like in *Protosciurus* and *Douglassciurus*. The peroneal process is most similar to that of *Spermophilus beecheyi* (*Emry & Thorington, 1982*), well-developed laterally, and in a rather anterior position. Overall, the calcaneus of *Paleosciurus* shows a few similarities with *Protosciurus* (ectal facet shape, tuber shape) which is considered to be arboreal according to *Korth & Samuels (2015)*. However, the more anterior peroneal process as well as the shorter and wider neck of *P. goti* show a more terrestrial lifestyle, in accordance with *Vianey-Liaud (1974)*, and our quantitative analysis suggests *P. goti* was also burrow-digging.

*Issiodoromys pauffiensis* is placed among the terrestrial generalists in the calcaneus analysis, and among the arboreal climbers in the astragalus analysis. This result is in disagreement with the findings of *Vianey-Liaud, Hautier & Marivaux (2015)*, which proposed that the species of the genus were jumping. However, it should be noted that no extant jumping taxa was correctly placed in the cross validated LDAs, which may explain the incoherence between those results. Qualitatively, the calcaneal and astragalar morphology of *I. pauffiensis* (Fig. 8B1 to 8B4 ; see also *Vianey-Liaud, Hautier & Marivaux, 2015*, Fig. 20.11) may be in accordance with a cursorial or jumping locomotion. The calcaneus is long, with a slender anterior part (Fig. 8B3). The peroneal process is in a distal-most position, and not very prominent, and the base of the ectal facet is perpendicular to the main axis of the bone (visible in lateral view). However, the calcaneo-cuboid facet has a unique shape (Fig. 8B4). Some qualitative characters of the astragalus of *I. pauffiensis* may explain why it is found in the climbing taxa. Indeed, the astragalar neck is long, and the trochlea is rather asymmetrical, similar to some extant climber taxa. However, the lengthening of the sustentacular facet, which joins distally with the AN facet, as well as the shape and curvature of the ectal facet are closer to that of cursorial and jumping taxa (e.g., Fig. 9).

Other skeletal elements presented in *Vianey-Liaud, Hautier & Marivaux (2015)*, are also in accordance with a jumping locomotion.

*Vianey-Liaud, Hautier & Marivaux (2015)* described the forelimb of *Pseudoltinomys gaillardi*, but were not able to make inferences about its locomotor repertoire. Our results (*P. gaillardi* is inferred as a fossorial species) disagree with the hypothesis of a "parallel adaptation of (the *Pseudoltinomys*) genus alongside that of *Issiodoromys*" (*Vianey-Liaud, Hautier & Marivaux, 2015*, Discussion and conclusion), since *Issiodoromys* comprises cursorial and jumping species rather than fossorial ones (see 'Results,' Figs. 6 and 7; (*Vianey-Liaud, Hautier & Marivaux, 2015*), Discussion and conclusion). The known elements of the forelimb (described in *Vianey-Liaud, Hautier & Marivaux, 2015*) do not exclude a fossorial or semi-fossorial lifestyle in this species. Using an "index of fossorial ability" (*Lagaria & Youlatos, 2006*), with the values given in *Vianey-Liaud, Hautier & Marivaux (2015)* (*Pseudoltinomys gaillardi*, Forelimb) the result found is $3.96/(23.42 - 3.96) = 0,203$. This value is similar to that of other extant fossorial taxa (*Lagaria & Youlatos, 2006*). Comparing with the values of the OLI index presented in *Samuels & Van Valkenburgh (2008)*, *P. gaillardi* is found to be in the range of semi-fossorial or semi-aquatic taxa. Those elements support our own result of a semi-fossorial lifestyle for *P. gaillardi*.

*Blainvillimys langei* and *Eucricetodon atavus* are both close to the fossorial group. *B. langei* was considered as a cursorial species by *Vianey-Liaud, Hautier & Marivaux (2015)*. However, looking qualitatively at the morphology of the calcaneus and astragalus (Fig. 8A; see also *Vianey-Liaud, Hautier & Marivaux, 2015*, Fig. 20.5), it does resemble other fossorial/semi-fossorial species. Indeed, the astragalar sustentacular facet is reduced and does not join the AN facet, whereas it usually is elongated antero-posteriorly in cursorial taxa (and in some species joins the AN facet, see Discussion, jumping and cursorial taxa; Fig. 9). Furthermore, the astragalar head is somewhat rounded, and the neck is long and slightly deflected medially, similar to other fossorial taxa (see Discussion—in other taxa, Figs. 3A and 4H for examples). The calcaneus (Figs. 8A3 and 8A4 , left) shows a wide (rather than elongated) anterior part, with a developed peroneal process, but in a more posterior position than in cursorial or jumping taxa (see Discussion, jumping and cursorial taxa ; Fig. 9).

It should be acknowledged that inferences based on the calcaneus seem more easily interpretable than the ones made on the astragalus, mainly because fossil specimens fall close or within the range of extant functional groups. The fact that the linear proportions of the calcaneus reflect its function could partly explain this result, whereas for the astragalus, the function may also be reflected by non-linear measurements such as the angle of the neck, the "roundedness" of the head, or the asymmetry of the trochlea. However, it can also be explained by the sampling of extant taxa, which is probably not representative of the full morphological diversity of extant rodents, nor representative of fossil morphological diversity.

## CONCLUSIONS

The morphologies of the calcaneus and astragalus are linked to the locomotion. Although rodents are generally very versatile in this regard, we showed that the bones of the ankle are good indicators of their main locomotor mode, particularly in specialized taxa (e.g.,

arboreal climbers, fossorial or jumping forms). This relation can be used, to some extent, to infer locomotor behaviours in fossil taxa. The astragalus and calcaneus are of particular interest in this regard, since they are often found in the fossil record. We showed that hypotheses can be made even based on isolated astragalus or calcaneus (e.g., *Candela & Picasso, 2008*; *Vianey-Liaud, Hautier & Marivaux, 2015*) in concert with other quantitative and/or qualitatitve analyses (*Samuels & Van Valkenburgh, 2008*; *Bover et al., 2010*). This type of approach has already been applied successfully in primates (e.g., *Dagosto, 1986*; *Dagosto, 1988*; *Dagosto, 1993*; *Gebo & Simons, 1987*; *Gebo, 1988*; *Gebo et al., 2001*; *Marivaux et al., 2010*; *Marivaux et al., 2012*). Although quantitative analyses have the potential to make inferences based on a large dataset, the results of this study suggest that these inferences should be made carefully, and with the largest possible sample size. Comparative anatomy, if possible accompanied by some functional interpretations, should still be used as a referential (rather than directly interpreting quantitative analyses), or as a mean to compensate for lack of morphological representativeness of the extant dataset.

Some important caveats are to be acknowledged when making inferences with this approach. First, since the shape of the bones does not respond to locomotion itself, but to the constraints that are applied to the bones, convergent morphologies may well occur between different types of locomotion that use similar movements. For instance, bipedal *versus* quadrupedal jumping would be hardly discernible among jumping species based on their astragalus and/or calcaneus solely. Similarly, semi-aquatic and cursorial taxa are hard to discriminate qualitatively or quantitatively because both groups use parasagittal movements of the foot to swim and/or run. Phylogenetic bias should also be taken into account, especially when phylogenetic groups are confounded with locomotor categories (e.g., in our case, most climber taxa are also sciurids). One should also be careful when using these "functional categories" (*Elissamburu & Vizcaíno, 2004*), which can be quite arbitrary.

Despite those limits, the study of the calcaneo-astragalar complex still provides some insights regarding the lifestyle of rodents. Even through comparative anatomy, arboreal climbers, terrestrial jumpers/cursors, and fossorial species can be discriminated relatively easily (*Candela & Picasso, 2008*, and discussion above). From a paleontological point of view, this type of comparative study represents a new source of behavioural and ecological data for fossil rodents, independent and comparable to other data, such as diet from dental categories (e.g., *Vianey-Liaud, 1991*) or tooth wear (e.g., *Gomes Rodrigues, Merceron & Viriot, 2009*), to improve the reconstructions of the ecology of extinct species.

## ACKNOWLEDGEMENTS

We thank the MNHN and UM collections for lending several specimens, and for allowing SG to visit their rodent collections ; François Catzeflis for lending the specimen of *Proechimys cuvieri* ; Thierry Noël and Marc Vianey-Liaud for lending some dissection tools; Camille Martinand-Mari and Emilie Liabeuf-Le Goff who offered access to the dissection room. We are also grateful to JX Samuels and one anonymous reviewer for their helpful comments on the manuscript.

## APPENDIX 1: ABBREVIATIONS (IN ALPHABETICAL ORDER)

| | |
|---|---|
| ABW | Astragalus Body Width |
| AmT | Astragalar-medial Tarsal |
| AmTL | Astragalar-medial Tarsal facet Length |
| ATL | Astragalus Total Length |
| ATW | Astragalus Total Width |
| CaCu | Calcaneo-Cuboid (facet) |
| CAL | Calcaneum Anterior Length |
| CCH | Calcaneum-Cuboid facet Height |
| CCW | Calcaneum-Cuboid facet Width |
| CPL | Calcaneum Posterior Length |
| CTL | Calcaneum Total Length |
| CTW | Calcaneum Total Width |
| EL | Ectal facet Length |
| EW | Ectal facet Width |
| HH | Head Height |
| HW | Head Width |
| ISEM | Institut des Sciences de l'Evolution de Montpellier |
| LBH/MBH | Lateral/Medial Body Height |
| lr/mr | lateral/medial radius |
| lTAH/mTAH | lateral/medial Trochlear Arc Height |
| LTL/MTL | Lateral/Medial Trochlear Length |
| MNHN | Muséum National d'Histoire Naturelle |
| MH | Maximum Height |
| NH | Neck Height |
| NL | Neck Length |
| NW | Neck Width |
| PPL | Peroneal Process Length |
| PPW | Peroneal Process Width |
| SL | Sustentacular Facet Length |
| SW | Sustentacular facet Width |
| TH | Tuber Height |
| TWc | Tuber width (calcaneus) |
| TWa | Trochlear Width (astragalus) |
| UAJ | Upper Ankle Joint |
| UM | Université de Montpellier |

## APPENDIX 2 : RESULTS FROM THE LDA ON THE ASTRAGALUS

Prior probabilities of groups:

| 1 | 2 | 3 | 4 | 5 | 6 |
|---|---|---|---|---|---|
| 0.08571429 | 0.17142857 | 0.25714286 | 0.11428571 | 0.25714286 | 0.11428571 |

Group means:

| | ABW | ATL | ATW | EL | EW | HH | HW |
|---|---|---|---|---|---|---|---|
| 1 | 0.2795295 | 0.7157586 | 0.3997427 | −0.04640842 | −0.2205371 | −0.2771638 | −0.11274398 |
| 2 | 0.3614328 | 0.6203164 | 0.4622937 | −0.01964534 | −0.2461056 | −0.3552226 | −0.09653397 |
| 3 | 0.2914743 | 0.6351321 | 0.5181896 | 0.01003638 | −0.2964897 | −0.4734017 | −0.07814465 |
| 4 | 0.3434863 | 0.6524569 | 0.5039599 | 0.01916780 | −0.2476862 | −0.3393500 | −0.04259003 |
| 5 | 0.2573425 | 0.6434438 | 0.4976781 | 0.01621850 | −0.2661444 | −0.3430905 | −0.10040117 |
| 6 | 0.3493795 | 0.5537950 | 0.4640582 | −0.07134248 | −0.1209143 | −0.5068228 | −0.10807614 |

| | LBH | LTL | MBH | MTL | NL | SL |
|---|---|---|---|---|---|---|
| 1 | −0.04828944 | 0.2035919 | −0.14182495 | 0.05016579 | 0.11808553 | 0.097273753 |
| 2 | −0.02811403 | 0.2269633 | −0.08059700 | 0.10305274 | −0.20574992 | 0.004098355 |
| 3 | −0.09020708 | 0.2322222 | −0.02890290 | 0.14349634 | −0.01280596 | −0.027459058 |
| 4 | −0.11295806 | 0.2015453 | −0.10303312 | −0.04200820 | 0.08437209 | −0.295533580 |
| 5 | −0.17564348 | 0.1981003 | −0.22247530 | −0.06470717 | 0.09343663 | −0.025417617 |
| 6 | −0.02876876 | 0.1446314 | −0.04667974 | 0.15404345 | −0.18764619 | 0.137110868 |

| | SW | TW |
|---|---|---|
| 1 | −0.8578186 | −0.15936139 |
| 2 | −0.6960436 | −0.05014519 |
| 3 | −0.7481649 | −0.07497487 |
| 4 | −0.5641562 | −0.05767271 |
| 5 | −0.5403148 | 0.03197464 |
| 6 | −0.7001367 | −0.03263137 |

Coefficients of linear discriminants:

| | LD1 | LD2 | LD3 | LD4 | LD5 |
|---|---|---|---|---|---|
| ABW | 0.5298125 | 8.0402184 | 17.0061985 | 1.3442958 | −0.1399471 |
| ATL | −8.8882780 | −5.6728674 | 6.6596415 | 0.7604116 | 11.5093998 |
| ATW | 7.5596580 | −1.5975077 | −8.9694737 | 3.6309800 | 1.4114182 |
| EL | 10.0707438 | −1.0919534 | −3.2055427 | −1.9495202 | −0.6091934 |
| EW | −4.2446584 | 2.1184284 | −3.3872345 | −5.3710509 | −2.4733596 |
| HH | −10.3904609 | 8.1813388 | −5.6211747 | 5.3682597 | −0.9179537 |
| HW | 9.1211196 | 0.8374204 | −3.4674061 | 1.4259092 | −2.1234192 |
| LBH | −7.5654175 | −1.5799693 | 7.4150531 | −0.1468658 | 0.7318520 |
| LTL | 7.0462574 | 8.8786241 | −3.6392035 | 8.2803044 | 8.4320446 |
| MBH | 4.2878351 | 3.3471524 | 6.7844033 | −2.2577348 | −0.4685301 |
| MTL | 7.2494378 | 2.7307732 | −7.5413673 | 0.3276344 | −0.8076582 |
| NL | 2.9710925 | 0.1671214 | −0.1328525 | 2.7764252 | −5.7927814 |
| SL | −1.8085936 | 5.8271209 | −5.2674725 | −1.7001297 | 1.0190663 |
| SW | −3.0465696 | −7.3689424 | 6.7211866 | −4.0141000 | 3.0599196 |
| Twa | −0.3733754 | −8.2921474 | −5.3921383 | −3.2060182 | 2.5812758 |

Proportion of trace:

| LD1 | LD2 | LD3 | LD4 | LD5 |
|---|---|---|---|---|
| 0.3632 | 0.3450 | 0.1557 | 0.1074 | 0.0288 |

## APPENDIX 3 : RESULTS FROM THE LDA ON THE CALCANEUS

Prior probabilities of groups:

| 1 | 2 | 3 | 4 | 5 | 6 |
|---|---|---|---|---|---|
| 0.08823529 | 0.17647059 | 0.26470588 | 0.11764706 | 0.23529412 | 0.11764706 |

Group means:

| | CAL | CCH | CCW | CPL | CTL | CTW | EL |
|---|---|---|---|---|---|---|---|
| 1 | 0.5428081 | −0.3241327 | −0.6434011 | 0.6768760 | 1.457412 | 0.3792084 | −0.27481128 |
| 2 | 0.4556747 | −0.2552781 | −0.5796393 | 0.6993746 | 1.368048 | 0.3452228 | −0.16196112 |
| 3 | 0.3780853 | −0.2766946 | −0.5022509 | 0.5913300 | 1.281944 | 0.4633359 | −0.05037334 |
| 4 | 0.2089128 | −0.1466701 | −0.3234480 | 0.4932873 | 1.255094 | 0.6599861 | −0.28739921 |
| 5 | 0.1651350 | −0.2302089 | −0.3435279 | 0.6051973 | 1.184352 | 0.6231515 | −0.04269456 |
| 6 | 0.2371690 | −0.2852894 | −0.4561087 | 0.8074916 | 1.266582 | 0.3870291 | −0.14270534 |

| | EW | MH | NH | NW | PPL | SL | SW |
|---|---|---|---|---|---|---|---|
| 1 | −0.6140210 | 0.4521675 | 0.10755490 | −0.3330313 | −0.4733468 | −0.2283696 | −0.8053747 |
| 2 | −0.5860735 | 0.3246363 | 0.09351646 | −0.3469678 | −0.3557590 | −0.3785312 | −0.5797817 |
| 3 | −0.4810829 | 0.2969979 | 0.11604921 | −0.3797099 | −0.3081778 | −0.5391912 | −0.6557702 |
| 4 | −0.6470726 | 0.1902021 | 0.01698257 | −0.3187746 | −0.1930455 | −0.5015808 | −0.5774419 |
| 5 | −0.5559576 | 0.2311915 | 0.11212594 | −0.5393386 | −0.2687376 | −0.2767090 | −0.5746003 |
| 6 | −0.4294913 | 0.3021705 | 0.12125905 | −0.4025320 | −0.4009309 | −0.2787967 | −0.6274753 |

| | TH | TW |
|---|---|---|
| 1 | 0.07867086 | 0.00179077 |
| 2 | 0.07368905 | −0.11617026 |
| 3 | 0.12681126 | −0.06130275 |
| 4 | 0.15181561 | 0.01915256 |
| 5 | 0.17980588 | −0.26918424 |
| 6 | 0.03055953 | −0.12893161 |

Coefficients of linear discriminants:

| | LD1 | LD2 | LD3 | LD4 | LD5 |
|------|------------|-------------|--------------|------------|------------|
| CAL | 6.4174449 | 3.50511872 | −7.07993865 | 0.6671641 | 2.2878551 |
| CCH | 7.3246441 | −4.73155266 | 7.61986282 | −6.9808590 | −3.0182532 |
| CCW | 4.5999642 | −9.36142862 | −0.30856777 | −5.9969305 | −2.3251114 |
| CPL | −0.4059243 | 4.04789287 | 0.04632179 | 1.0338494 | 0.5343113 |
| CTL | −23.6936015 | −7.35723085 | 16.71228120 | 1.2877966 | −4.8971000 |
| CTW | 4.5413249 | 12.32735358 | −5.17462522 | 2.3889705 | −3.3709005 |
| EL | 0.2910011 | −4.44676585 | −2.42225487 | −4.1516248 | 1.0831386 |
| EW | 3.8885261 | −4.11006743 | 1.66413085 | −2.0640945 | −3.1014178 |
| MH | −0.7069701 | 5.14917621 | −12.26742443 | 2.5938795 | 3.9900014 |
| NH | −2.0762885 | −1.20067995 | 11.50849609 | −17.4323781 | −2.7658429 |
| NW | −1.2764186 | −3.97425246 | 1.75307936 | −4.2123723 | −0.2883815 |
| PPL | −2.9196060 | 1.92081145 | 2.05688635 | 4.6565381 | 2.5390342 |
| SL | −5.6253588 | 4.35642986 | 0.16891955 | −1.4427326 | −3.4376052 |
| SW | −2.2266841 | 0.05400985 | 4.30230555 | 3.8414218 | 3.5808466 |
| TH | −0.9866944 | −1.85309037 | −3.43513075 | 7.7087568 | 3.4236471 |
| Twc | −0.3359066 | −11.67277792 | −3.74125392 | −1.6726337 | −4.5590015 |

Proportion of trace:

| LD1 | LD2 | LD3 | LD4 | LD5 |
|--------|--------|--------|--------|--------|
| 0.5206 | 0.2993 | 0.0758 | 0.0592 | 0.0451 |

### Funding
The authors received no funding for this work.

### Competing Interests
The authors declare there are no competing interests.

### Author Contributions
- Samuel Ginot conceived and designed the experiments, performed the experiments, analyzed the data, contributed reagents/materials/analysis tools, wrote the paper, prepared figures and/or tables, reviewed drafts of the paper.
- Lionel Hautier conceived and designed the experiments, wrote the paper, reviewed drafts of the paper.
- Laurent Marivaux conceived and designed the experiments, wrote the paper, reviewed drafts of the paper.
- Monique Vianey-Liaud conceived and designed the experiments, contributed reagents/materials/analysis tools, wrote the paper, prepared figures and/or tables, reviewed drafts of the paper.
## Data Availability

The raw data has been supplied as Supplementary File.

## Supplemental Information

Supplemental information for this article can be found online at http://dx.doi.org/10.7717/peerj.2393#supplemental-information.

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
