# Peer review of "Ecomorphological analysis of the astragalo-calcaneal complex in rodents and inferences of locomotor behaviours in extinct rodent species"

_PeerJ, doi:10.7717/peerj.2393_

## Round 0.1 · original submission · Major Revisions

I have received two reviews of your paper. Both reviewers consider that your manuscript is interesting and makes a significant contribution to understanding rodent ecomorphology. I am particularly concerned in relation to the sample data (please give details of the number of specimens and their identification) and the locomotory categories used. Differences between locomotor modes and habitat use must be clearly stressed. Re-categorization of the taxa in relation to their locomotor behavior seems to be the best option.

I would like to see a revised version of your manuscript that takes a point by point account of the comments of the reviewers.

Reviewer 1 ·

Basic reporting

No Comments

Experimental design

No Comments

Validity of the findings

No Comments

Additional comments

Comments to the Author
This manuscript examines the relationship between the morphology of the astragalus and calcaneus and a wide range of locomotor habits in three main groups of rodents (Sciuroidea,Myodonta, and Ctenohystrica). These are a species rich and ecologically diverse group that warrants further study, both in terms of ecological adaptations and evolutionary diversification. Detailed quantitative examination of the relationship between morphology and ecology in a broad sample of taxa is particularly important, and this study will likely be of use to a broad spectrum of scientists.
I found the study to be interesting and enjoyed reading the manuscript. The introduction does a good job introducing the group and the objectives of this study. The methods used were appropriate for this type of study, but more detail is needed on the samples used and the locomotives categories. In general, the results and discussion were good presented but there are a few portions that would be more appropriately presented in another part of the manuscript. The figures were particularly well done. This manuscript improves our understanding of the relationship between morphology and ecology in rodent and can help improve our understanding of their evolutionary history.
My comments are summarized below.

Material and Methods
Line 84-96: This section could be described in substantially greater detail. First paragraph, could you please give some precision about the sample composition, (number of specimens, species and genera). All adult individuals?. Please indicate the authors that followed for osteological nomenclature.
Line 117-130: Locomotory categories. I found the categories problematic. Some of these categories describe locomotor behavior or functional movements (“jumping”, “fossorial”), while others appear to just describe habitat (e.g. “terrestrial”, “arboreal”). I recommend that the taxa be recategorized with respect to locomotor behavior or behaviors that are likely to be associated directly with postcranial adaptations (e.g. jumping, quadrupedalism, climbing, digging, etc) not habitat. This makes more sense in terms of the stated goals of the paper.
It is not clear to me why you have used the category semi-acuatic in quantitative analysis and not in qualitative analyses. If one of the purpose of the paper is to describe qualitatively both bones to define character states that best characterize the motion range of the ankle, and the favoured locomotor mode, then this point can not apply for the category semi-acuatic because they did not osteological description.
Also, I would suggest including the definition of locomotor modes.
Line 131-139. Why you have include11 species (16 specimenes) in qualitative analyses and 35 species (56 specimenes) in quantitative analysis?. If they all have astragalus and calcaneus why not provide information of the other genera in the qualitative analyses?. Perhaps you can include more representatives of each family and see better variation. If the idea was to include genera that represent the different locomotor types, please indicate this in the text of manuscript.
Please indicate the authors that followed for measurement of astragalus and the calcaneus. Why these measurements were selected? (e.g. Importance functional). Perhaps the authors should add a table in the text with the osteological measurements.

Result
Line 166: Delete of the title "of dissected specimens"
Line 173-176: This information is more appropriately included as a portion of the Material and Methods.
Osteological descriptions: This part is long. Perhaps authors can make a general description of the astragalus and calcaneus by family and not by species. For example, make a general description of the structures of astragalus indicating the differences between species (e.g. the sustentacular and ectal facets are separated by wide sulcus, except in .....).
There are comments along the descriptions that should be presented in discussions. For example: line 250- 257; line 274-280; line 397-399; line 401-403.
Quantitative analyses: I think they should only indicate how differed the groups and which were the variables that define it.
I don’t understand why the authors explaining all variables in the result. I think this should be presented in functional morphology analysis in the discussion.
Line 624-626. This information is more appropriately included as a portion of the Material and Methods.

Discussion
This section is long. There is information already been mentioned in the results. Please delete the redundant paragraphs.
Line 806: Title should be: Functional analyses of locomotor types.

Table 1: The caption should also be revised, perhaps as follows: " Rodents species examined in this study, locomotory category, and literature sources consulted." I would recommend delete the column of the collection, specimen, description, dissection and LDA.
Figure 1: This caption should be revised to reflect only the main structures of the bone, rather than the area of insertion of muscle. This manuscript is about osteology.

·

Basic reporting

No Comments

Experimental design

No Comments

Validity of the findings

No comments

Additional comments

The manuscript by Ginot et al. examines the morphology of the astragalus and calcaneum in a variety of rodents, from several distinct clades and with varying locomotor ecology. The manuscript was well written overall, the introduction provides the necessary background for the study and the methods use a combination of qualitative dissections and quantitative analyses appropriate for characterization of how calcaneo-astragalar morphology reflects locomotion. The results are clearly written, the figures are well done, and the discussion puts the study in a broader context. This study makes a significant contribution to understanding rodent ecomorphology and will likely be of use to a broad group of readers, including anatomists, paleontologists, and evolutionary biologists. I had previously reviewed this manuscript and find that the authors have adequately addressed all of the points I raised in that review.

A few suggestions for improving the manuscript follow:
1. Page 9, line 180: The subfamily and tribe information are included for other members of the Sciuridae, but are missing for Marmota.
2. Page 11, lines 236 - 239: These comments would be more appropriately placed on page 9, following the heading for the Sciuridae.
3. Page 20, line 514: "Erethizontid" should not be capitalized.
4. Page 23, line 601: The quantitative analysis results should start with a more generalized lead in statement. Perhaps indicating that the LDA successfully separated locomotor groups based on astragalus and calcaneum morphology (report Wilk's lambda of LDA, probability, and refer to figures 6 and 7). As currently written, it somewhat jarringly jumps to describing individual groups.
5. Page 24, lines 634, 636: These lines use the term "runner", while "cursorial" is used more throughout the manuscript text, and in other literature sources.
6. Pages 30 and 32, lines 808 and 899: The headings for these locomotor groups should not be in bold, like the others.
7. Page 36, line 1018: It should be noted how the coypu is distinct from the beaver and capybara. Part of the reason Myocastor does not group with the other semi-aquatic taxa in Figures 6 and 7 may be due its smaller body size. The functional demands of swimming (i.e. overcoming drag) become more pronounced at larger size, requiring greater specialization. As Hydrochoerus and Castor are the largest extant rodents, their calcaneo-astragalar specialization for swimming should not be surprising.
8. Page 40, lines 1137, 1143: The species name for I. pauffiensis should not be capitalized.
9. The format of some citations should be checked. For example on Page 43, line 1236 the citation of Antione et al. includes an ellipsis "..." rather than listing all authors, should that be the case? SImilarly on Page 48, line 1390 the article title has capitalized words throughout, unlike many other citations. Many of the citations have alignment and spacing issues.
10. Page 57, Table 1: All species names should be italicized.

---

## Round 0.2 · accepted · Accept

Thank you very much for your consideration of the suggestions of our reviewers. Note that a few more things have been suggested to improve your manuscript which can be dealt with while in production.

Reviewer 1 ·

Basic reporting

No comments

Experimental design

No comments

Validity of the findings

No comments

Additional comments

I could see that the authors worked to incorporate many of the suggestions and criticisms in this new version of the manuscript and I consider that it was substantially improved. In general, I am quite pleased with changes performed in the manuscript. I think the Osteological descriptions are clearly described now, as well as the discussion.

Small suggestions for improving the manuscript follow:

Results
Line 604: add " r" to cusorial.

Discussion
Line 696: delete "u" and add "n" to ruuning/paddling
I think that authors should decrease the number of titles in the discussion. As a suggestion, this two titles: Quantitative differences between locomotor groups (or Functional Morphology of the astragalar and calcaneal Features) and Functional analysis of locomotor types